# Practice beyond performance stabilization increases the use of online adjustments to unpredictable perturbations in an interceptive task

Crislaine Rangel Couto[1]☯, Cláudio Manoel Ferreira Leite[2]☯,
Carlos Eduardo Campos[1]☯, Leonardo Luiz Portes[3]☯, Cíntia de Oliveira Matos[1]‡,
Suziane Peixoto Santos[4]‡, Natália Fontes Alves Ambrósio[5]‡, Hani Camille Yehia[6]‡,
Herbert Ugrinowitsch[1]☯*

**1** Sports Department, Universidade Federal de Minas Gerais-UFMG, Belo Horizonte, Minas Gerais, Brazil,
**2** Department of Physical Education and Health Science, Universidade Federal de São João del-Rei-UFSJ, São João del-Rei, Minas Gerais, Brazil, **3** Department of Mathematics and Statistics, University of Western, Perth, Western Australia, Australia, **4** Department of Sports Science, Universidade Federal do Triângulo Mineiro-UFTM, Uberaba, Minas Gerais, Brazil, **5** Department of Prevention and Health Promotion, Centro Universitário Una, Belo Horizonte, Minas Gerais, Brazil, **6** Department of Electronic Engineering, Universidade Federal de Minas Gerais-UFMG, Belo Horizonte, Minas Gerais, Brazil

☯ These authors contributed equally to this work.
‡ CdeOM, SPS, NFAAand HCY also contributed equally to this work.
* herbertu@ufmg.br

## Abstract

In recent decades, research has focused on motor adjustments in interception tasks within predictable environments. However, emerging studies suggest that continued practice beyond performance stabilization enhances the ability to adapt to unpredictable events. The objective of this study was to investigate the effects of practicing until performance stabilization versus extended practice through superstabilization on the ability to adjust to unpredictable perturbations in intercepting a moving target. We hypothesized superstabilization would better facilitate motor adjustments in response to unpredictable perturbations. Forty participants engaged in an interception task until they achieved either performance stabilization or superstabilization. Subsequently, both stabilization and superstabilization groups were tested in an unpredictable environment, where, in certain trials, the target's velocity unexpectedly changed after the onset of the movement. The findings revealed that the superstabilization group made more adjustments, showing more number of corrections (N-cor), in response to these perturbations than the stabilization group, attributed to their developed capacity to use online feedback as a control mechanism more efficiently. In contrast, the practice until performance stabilization did not foster this adaptive mechanism. These results support the notion that learning is a dynamic process that extends beyond the point of performance stabilization, emphasizing the benefits of continued practice for mastering motor tasks in variable contexts.

**Data availability statement:** All relevant data are Supporting information files of manuscript.

**Funding:** Authors "CRC, CEC, LLP, COM e NAFA" received schoolarship from Coordenação de Aperfeiçoamento de Pessoal de Nível Superior – Brasil (CAPES) – Financing Code 001. https://www.gov.br/capes/pt-br Author "HU" received research support from Fundação de Amparo à Pesquisa do Estado de Minas Gerais (FAPEMIG) PPM IX, PPM-00773-15. http://www.fapemig.br/pt/ The authors "CMFL, SPS e HCY" did not receive any research support from any company. The funders had no role in study design, data collection and analysis, publication decisions, or manuscript preparation.

**Competing interests:** The authors have declared that no competing interests exist including any financial, personal, or other relationships with other people or organizations.

## Introduction

Hitting a ball during a tennis game requires intercepting a shot from the opposite court. Interception actions of moving targets clearly reveal distinct abilities to solve motor problems due to different skill levels [1]. These differences in skill level become even more apparent when the task is performed in complex and dynamic contexts [2] characterized by environmental changes named perturbations and that require adjustments in an action [3].

Perturbations are a constant in motor performance (e.g., a ball that changes speed after touching the floor) and impose specific sensorimotor demands requiring specific motor adjustments. These perturbations can be predictable or unpredictable to the performer [4,5]. Predictable perturbations can be anticipated, consequently, the action planning may contain the changes necessary to adapt to the perturbation. Conversely, unpredictable perturbations cannot be anticipated, consequently, the action is modified in response to sensorial feedback during the execution. In essence, unpredictable perturbations in interception tasks necessitate real-time, online corrections [6].

Many aspects of motor learning influence the ability to perform online corrections [7] in an attempt to maintain performance accuracy under perturbations [8,9]. Previous studies have shown that the level of performance stabilization attained during the learning phase (prior the insertion of perturbations) is a aspect that influences motor control and performance when facing unpredictable perturbations [3,9]. Experimentally, different levels of performance stabilization were established based on pilot studies designed to assess how volunteers perform different tasks and identify specific stabilization criteria to each task. Over the past two decades, two levels of performance stabilization have been manipulated in studies of motor learning and control: stabilization and superstabilization levels [3,9–11].

Performance is considered stable when an action is consistently reproduced within a small bandwidth of error and with the movement spatiotemporal standardization, resulting in correct responses in consecutive trials during the learning phase [5,12]. This rationale has been applied in studies involving complex coincident-timing and isometric force control tasks. In a complex coincident-timing task, performance is considered stable when the error is equal to or less than 30 ms, and this level of performance is repeated for three trials in a row [3,5,10,13]. Furthermore, in an isometric force control task [9], it was found that participants could perform no more than four consecutive trials with an error smaller than or equal to 5%, and this was adopted as a criteria for performance stabilization in this task.

Superstabilization level means that the practice goes beyond stabilization, requiring a distinct criterion from that used for the stabilization level. Operationally, the superstabilization level criteria is defined as achieving the stabilization criterion multiple times. For instance, Fonseca et al., 2012 [3]; Ugrinowitsch et al., 2011 [10]; Couto et al., 2021 [11] and Campos et al., 2022 [13] adopted the approach of repeating the stabilization criteria over six consecutive or non-consecutive blocks during the learning phase. Overall, studies have shown that, under constant practice conditions,

achieving the superstabilization level (i.e., six blocks of three or four successful trials in a row, depending on the task) can better deal with perturbations than practicing only until stabilization [3,9,14,15].

In this context, Ugrinowitsch et al., 2014 [14] and Couto et al., 2021 [11] have shown that, although the stabilization and superstabilization groups achieved simillar accuracy at the end of the learning phase, the superstabilization group finished the phase with higher performance variability than the stabilization. However, other studies failed in finding to find differences in performance variability when comparing stabilization with superstabilization [16], despite the superstabilization group demonstrating higher performance accuracy when unpredictable perturbations were inserted. One possible explanation for the lack of difference in performance variability between levels is that variability differences can be found only in motor control measures rather than in performance. The latter depends on the characteristics of the task as well, and the studies that did not find differences did not assess motor control variability [17].

Higher variability during the learning phase may be related to the use of online corrections and consequently the best performance accuracy when unpredictable perturbations are inserted [13] even though no apparent difference in performance accuracy occurs at the end of the practice phase [3,10]. Therefore, one possible explanation for superiority of the superstabilization level in face to unpredictable perturbations is that the higher variability in performance and/or motor control seems to allows sensorimotor system's high adaptability [3,12,16].

Over a constant schedule, the practice until performance stabilization (for example, one block of three consecutive correct trials in a complex coincident task), helps build a robust and consistent internal representation of movement dynamics, named internal models. Internal models contain information concerning the characteristics of the limbs and environmental dynamics involved in the task [17,18] allow the performer to use the pre-program mechanism to control the action [12,19]. However, this pre-programmed mechanism may prove inefficient when faced with an unpredictable perturbation. In contrast, even over a constant practice schedule, the practice that extends beyond performance stabilization, named superstabilization [3,10] appears to increase the internal model's ability to change [3,18]. It allows performers to utilize feedback control mechanisms for making online adjustments.

Internal models enable efficient motor control by facilitating the extraction and utilization of task-relevant information to generate appropriate adjustments [20,21]. For example, in coincident timing tasks, expert baseball athletes present a remarkable adaptive capacity in the face of changes in stimulus speed thanks to sophisticated speed detection and response adjustment strategies [21]. Although coincident timing tasks provide an approximation to interceptive actions as these types of tasks present perceptual similarities [22], interceptive actions show some perceptual-motor particularities that require specific consideration, such as an actual approximation of the effector to the moving target and a clear hit/miss condition [23–26] that might influence the mechanisms of control.

The mechanism of control in interceptive motor tasks can be observed from the kinematics analysis, such as time to peak velocity and the number of phases in the acceleration curve of the action [23,27,28]. A peak velocity that coincides with or close to the moment of interception, along with a monophasic acceleration curve, typically indicates a predominance of pre-programming mechanism [22,29]. In these cases, the acceleration curve (velocity derivative) presents a single positive component (i.e., monophasic profile). The same monophasic shape of the acceleration curve has been found when the target travels at a constant speed and predictably changes velocity before the movement onset [24,25]. Similar results were found when the target accelerates or decelerates [25].

When the practice of an interception task is under a predictable context, a constant practice schedule, the pre-programming mechanism predominates [11]. In such conditions, the motor control system can predict the outcomes of actions based on sensory information gleaned from previous practice trials [30,31]. On the other hand an unstable context demands a feedback control mechanism, which adjusts the function to context changes [32,33]. Furthermore, the successful interception of a moving target demands great prediction and planning to allow sufficient time for adjustments when necessary [34]. For example, Fialho and Tresilian [25] showed that in interceptive tasks, even when the movement

time is between 130 and 170 ms regardless of the target velocity, all acceleration curves presented a monophasic profile, demonstrating the need for more time for any adjustment by the motor control system.

In unpredictable contexts, when the target's speed changes after the onset of the movement, maintaining or achieving proper performance levels requires the use of online feedback mechanisms of control [35]. Under these conditions, the peak of velocity occurs earlier, and the velocity curve presents inflections indicating movement adjustments. For example, Tresilian and Plooy [23] found valleys in the acceleration curve (i.e., bi or polyphasic profile according to the number of valleys), signaling corrective sub-movements during an interception task. The efficiency of the online feedback mechanism in rapid movements hinges on swift corrections to the motor command [36] and seems to be linked to the level of performance stabilization [3,18]. Although many daily activities and sports involve interception, and the pursuit of success in such tasks is widespread [1], to our knowledge, the relation between the level of performance stabilization and the mechanisms underlying adjustments to unpredictable perturbations in interceptive actions has yet to be thoroughly investigated.

The purpose of the present study is to investigate the effects of two levels of performance stabilization on the adjustments to unpredictable perturbations in the interception of a moving target. We adopted a virtual interceptive task to manipulate two levels of performance stabilization during the learning phase, followed by a testing phase with unpredictable perturbations interspersed with control trials, i.e., the same condition as the learning phase.

We assumed that practice beyond performance stabilization would enhance online adjustments to unpredictable perturbations, as evidenced by the movement's kinematics observed on an increased number of phase valleys in the acceleration curve and a reduced time to reach peak velocity. This will lead to improved performance under unpredictable conditions.

## Materials and methods

### Participants

Forty-two young university students (26.02 ± 2.02 yr.; 22 men), self-declared right-handed, with a normal or corrected-to-normal vision, no history of neurological impairment or orthopedic limitations of the upper limbs, and inexperienced in the task participated in this study. This sample size was determined by using GPower (version 3.1.2; Franz Faul, Universitat Kiel, Germany) [37]. The following parameters were used to determine the sample size: a power of 0.80 and an expected effect size of 0.3. Our sample size was set to $N = 18$, but a larger sample size was used because this could increase the reliability of our results. All participants provided written informed consent prior to testing and were informed that they could withdraw their consent at any time. The local ethics committee approved this study (n. 30544714.7.000.5149), and the procedures followed the ethical standards in the Declaration of Helsinki 1964, amended in 1989.

### Instruments and task

The setup for the virtual interception task (Fig 1) consisted of an Intel Celeron 2.20 GHz computer, a 17" monitor (Dell 60 Hz, 1366x768'), a 35 cm long wireless drawing tablet (WACON- INTUOS 3–9 x 12) with a capture frequency of 200 Hz, and a digital pen (INTUOS 3). A foaming (EVA) plate 2 cm thick with a posterior-anterior cut in its center (27.7 cm long and 1.7 cm wide) was landed on the tablet, forming a groove constraining the digital pen movement to 1 degree of freedom (i.e., forward). Moreover, the task was projected (SONY VLP ES7® with 60 Hz) onto a 304 cm wide and 228 cm high white screen placed 370 cm in front of the participant. The Virtual Interception Task (VIT), developed in Labview® (Leonardo Portes; Crislaine Rangel Couto & Herbert Ugrinowitsch, UFMG, Belo Horizonte/MG, Brazil), controlled the interception task, data acquisition, and processing. The VIT synchronized the virtual task (target and effector) and the tablet and controlled the target velocities.

The Interception Task (IT) was to hit a virtual target (4 x 6 cm yellow rectangle) that moved along a horizontal rail 304 cm long, both projected on the screen. To initiate every trial, the participant positioned the pen in the proximal part of the groove (closer to the body), maintaining the elbow joint at 90°. The yellow virtual target appeared on the right side of

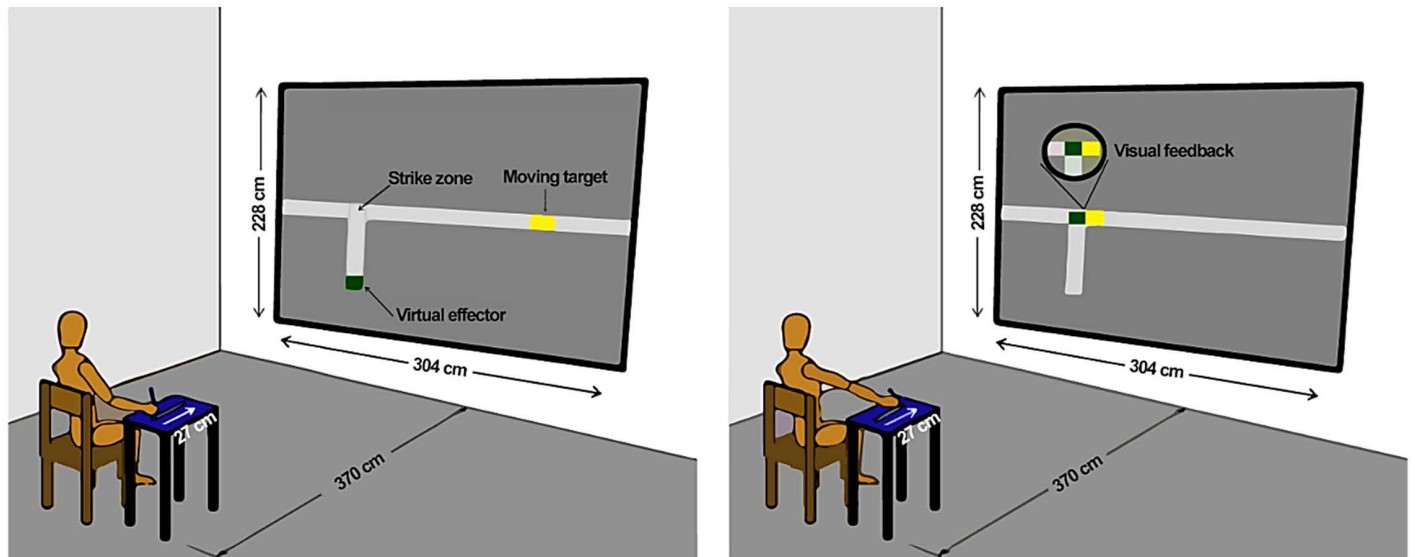

**Fig 1. The virtual target (A) moved at a velocity controlled by VIT, over 213 cm to reach the strike zone (B).** To control the virtual effector to reach the strike zone at the same moment as the target, the participant moved a pen over the tablet within 200 to 250 ms.

the screen, and the experimenter controlled the moment the target started traveling through the rail to the left end of the rail, which ranged from 1 to 3 s. From its initial appearance to the center of the strike zone, the target traveled 213 cm. When the target was near the interception area, the participant had to move the digital pen along the groove (shoulder flexion and elbow extension). The digital pen controlled the screen's virtual effector (2 x 4 cm green rectangle), which moved along the vertically projected rail, which was exactly the same length and width as the physical rail. The participants had to move the virtual green effector and intercept the yellow target within the width of the target and inside the strike zone, which corresponded to intercepting the target within a 5 cm bandwidth relative to its center.

### Procedure and design

Every participant received demonstrations and plain explanations about the task. Then, he/she sat on a chair with the digital table on the right-hand side. The chair was regulated on the horizontal and vertical axis to maintain the elbow joint at 90º about the shoulder. The velocity at which the target moved was 145 cm/s. A 4 × 6 cm target moving at 145 cm/s and a 2 × 4 cm effector resulted in a 68,96 ms time window. This time window represented the period for which contact between the effector and a moving target was possible in the strike zone [38].

The 42 participants were balanced by sex and randomly assigned into two groups previously prepared according to the level of stabilization required for the experiment: a Stabilization Group (SG) and a Superstabilization Group (SSG). This preparation consisted of performing the IT at a constant speed (145 cm/s) until they reached a group-specific performance criterion related to intercepting the target. The Stabilization Group practiced until they intercepted the target three trials in a row. The Superstabilization Group practiced until they reached the same performance criterion (three right trials in a row) across six blocks throughout the phase, which differentiated the Stabilization and Superstabilization groups. As such, the two groups represented two different levels of learning and guaranteed the manipulation of our independent variable. The SG practiced the task until reaching the performance criterion with an error smaller than or equal to 5 cm. The error was calculated by comparing the distance of the effector's center to the target's center. The SSG practiced the task until reaching its performance criterion with the same error bandwidth of 5 cm. In the SG, participants were required to reach

the performance stabilization criterion (one block of three attempts) over a maximum of 200 attempts. In the SSG, the participants had to reach its criterion (six blocks of three attempts) throughout the learning phase, with a maximum of 320 attempts. In both groups, participants who failed to reach the criterion within the maximum number of attempts established in the pilot study were excluded from the sample. This criterion follows the procedures of previous studies when two different levels of stabilization showed no difference in performance accuracy when the visual stimulus ran at a constant velocity [3,10]. Two participants failed to meet the criterion of superstabilization performance over 320 trials and were not included in data analysis.

Participants were instructed to perform a fast movement with a duration between 200 and 250 ms. This duration was a control variable and allowed the participant to pre-program their movements and use online feedback control [23] when necessary, especially in face to perturbation. Thus, considering the distance of 27.7 cm for participants to move the pen along the physical rail, the movement speed of his right upper limb should be between 110.8 and 135 cm/s. To keep movement duration within the specified range, we provided qualitative verbal information about it after every trial in the following way: (a) Movement times (MTs) below 179 ms – "the movement was very fast;" (b) MTs between 180 and 199 ms - "the movement was fast;"(c) MTs between 200–250 ms - "good movement time;" (d) MTs between 251–270 ms - "the movement was slow;" and (e) MTs over 271 ms - "the movement was very slow". With regards to feedback on target interception, the IT projected an image of the green effector and the yellow target at the moment the effector crossed the interception area, which served as a post-trial visual knowledge of results (KR). The participant could request a rest break at any time during the preparation.

After reaching the performance criterion, there was a 10-minute rest interval before the Exposure phase commenced. The Exposure phase consisted of 129 trials. In 111 of the 129 trials, the target traveled at the same velocity as the Learning phase (i.e., 145 cm/s), corresponding to a time window of 68,96 ms. In the other 18 trials, the stimulus started moving at 145 cm/s, but the velocity changed immediately after the onset of the interception movement. In nine trials, these changes randomly increased to 200 cm/s named "Perturbation fast" ($P_{fast}$), and in the other nine trials, the velocity decreased to 90 cm/s, named "Perturbation slow" ($P_{slow}$). The time window of $P_{fast}$ was 50 ms, and the time window of $P_{slow}$ was 111 ms. Previous studies have used the same system for organizing control trials and trials with perturbation. [3,13]. The participants were instructed that the target velocity would change, but they did not know how or in which trials would occur, making perturbations unpredictable (Fig 2).

## Measures and data analysis

The number of corrective sub-movements (N-cor), considered as the number of inflections (valleys) in the acceleration curve [23,39]. Valleys in the acceleration curve indicated corrective sub-movements when they reached at least 2% of the magnitude of the previous acceleration peak [23]. The relative time to peak velocity (tPV%) refers to the proportion of the total movement time required to reach peak velocity [40]. The N-cor and the tPV% indicated the mechanisms and strategies of control. The constant error (CE), expressed as the magnitude and direction of deviation from the center of the target in cm indicated performance accuracy [41]. Constant error (CE) was calculated as the distance (cm) between the center of the target and the center of the effector when the latter reached the strike zone.

To analyze the number of corrections (N-cor), we considered the average number of corrections from the trials with either perturbation. We separated perturbation fast ($P_{fast}$) and perturbation slow ($P_{slow}$), them we applied t-Student tests to independent samples to compare the average of the number of corrections performed by SG and SSG during each perturbation (i.e., $P_{fast}$ and $P_{slow}$). The separation of fast and slow perturbations is important because $P_{fast}$ and $P_{slow}$ should cause opposite error bias.

For the analysis related to the relative time to peak velocity (tPV%) and constant error (CE) data, we first separated perturbation fast ($P_{fast}$) and perturbation slow ($P_{slow}$) and their respective and immediately preceding (Pre) and following (Post) trials. Both perturbations ($P_{fast}$ and $P_{slow}$) were organized into three moments throughout the Exposure phase: Early,

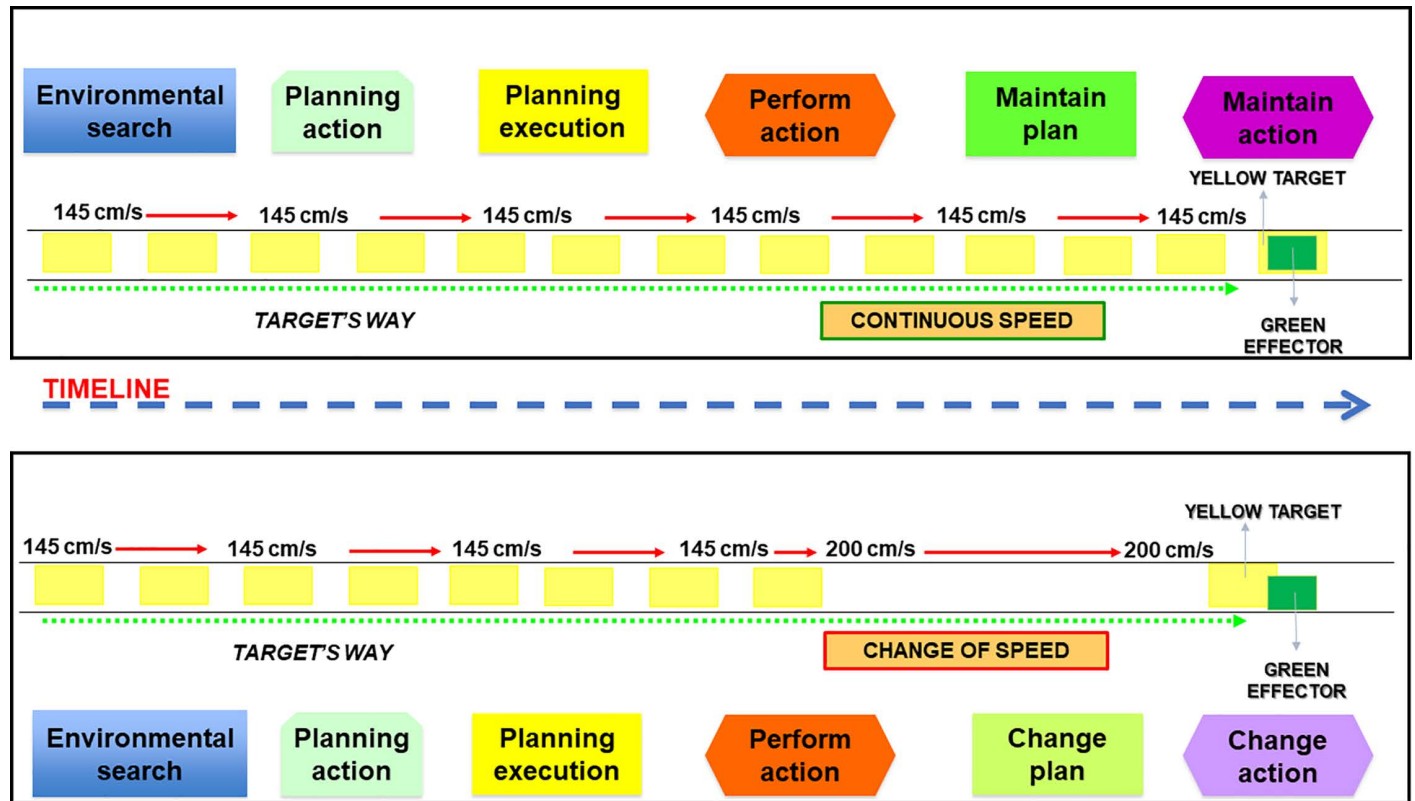

**Fig 2. Simulation of the experimental protocol during the Exposure phase.** The upper part shows a control trial when the target starts moving in the direction of the strike zone at constant velocity. Firstly, the participant searches for environmental information, plans the action, executes the plan, and starts the action. Based on the same target velocity, the participant maintains the action plan and the action. The bottom part shows a perturbation trial when the target changes velocity after the onset of the movement. Firstly, the participant searches for environmental information, plans the action, executes the plan, and starts the action. Based on the change in target velocity, the participant changes the action plan and the action trying to reach the target.

Intermediate, and Latter Moments. Each moment was composed of the Pre, $P_{fast}$ or $P_{slow}$, and Post blocks, each containing the averages of three trials. Therefore, the Early Moment contains the average of the first three perturbation trials (1–3) and the average of their respective Pre and Post control trials. This same organization was used for the Intermediate Moment, containing the average of the fourth, fifth, and sixth perturbation trials (4–6) and the average of their respective Pre and Post control trials. Finally, the same organization was maintained for the Later Moment, containing the average of the seventh, eighth, and ninth perturbation trials (7–9) and the average of their respective Pre and Post trials, i.e., control trials. Based on this organization, for the fast perturbation, the relative time to peak velocity (tPV%) and constant error (CE) data were described as follows:

Early Moment: Pre, P1-3$_{fast}$, Post

Intermediate Moment: Pre, P4-6$_{fast}$, Post

Later Moment: Pre, P7-9$_{fast}$, Post

Based on this organization, for the slow perturbation, the relative time to peak velocity (tPV%) and constant error (CE) data were described as follows:

Early Moment: Pre, P1-3$_{slow}$, Post

Intermediate Moment: Pre, P4-6$_{slow}$, Post

Later Moment: Pre, P7-9$_{slow}$, Post

The CE and tPV% measures from each of the exposure phase moments (Early, Intermediate and Later) and perturbations (fast and slow) were analysed using a two-way mixed-design ANOVA consisting of groups (SG and SSG) and blocks (Pre, P, Post) factors.

The significance level was set at $p < 0.05$ for all inferential statistics, and Tukey's post hoc test was used for pairwise comparisons. Additionally, to examine the magnitude of the effects for each analysis, we used partial eta squared ($\eta_p^2$). Effects were considered large when $\eta_p^2$ was above 0.14; medium for $\eta_p^2$ between 0.06 and 0.14; small for $\eta_p^2$ between 0.01 and 0.06; and trivial when $\eta_p^2$ was below 0.01 [42]. Preliminary data analyses indicated that all data fit normality (Shapiro-Wilk's test), ($p > 0.07$).

## Results

### Fast perturbations

**Early Moment: Pre P x P1-3$_{fast}$ x Post P.**  The analysis of the number of corrections on the first moment of P$_{fast}$ (P1-3$_{fast}$) shown in Fig 3 indicates that SSG performed more corrections than SG ($p = 0.03$). The analysis of tPV (%) did not show difference between groups, $F(1, 38) = 0.27$, $p = 0.60$, $\eta_p^2 = 0.007$, power (1-β) = 0.08; blocks, $F(2, 76) = 0.61$, $p = 0.54$, $\eta_p^2 = 0.01$, power (1-β) = 0.14; nor interaction, $F(2, 76) = 1.23$, $p = 0.29$, $\eta_p^2 = 0.03$, power (1-β) = 0.26. The analysis of CE showed difference between blocks, $F(2, 76) = 98,56$, $p = 0.001$, $\eta_p^2 = 0.72$, power (1-β) = 1.00. The post hoc detected that the CE increased (i.e., later) when the perturbation was inserted (P1-3$_{fast}$) compared to Pre ($p < 0.05$). With the withdraw of perturbation (Post), the CE decreased (i.e., earlier) compared to P1-3$_{fast}$ ($p < 0.05$). There was no significant difference between groups, $F(1, 38) = 0.91$, $p = 0.76$, $\eta_p^2 = 0.002$, power (1-β) = 0.06 and nor interaction, $F(2, 76) = 2.69$, $p = 0.07$, $\eta_p^2 = 0.06$, power (1-β) = 0.51.

**Intermediate Moment: Pre P x P4-6$_{fast}$ x Post P.**  The analysis of the number of corrections on the second moment of P$_{fast}$ (P4-6$_{fast}$) showed that SSG performed more corrections than SG ($p = 0.001$). The analysis of tPV (%) showed

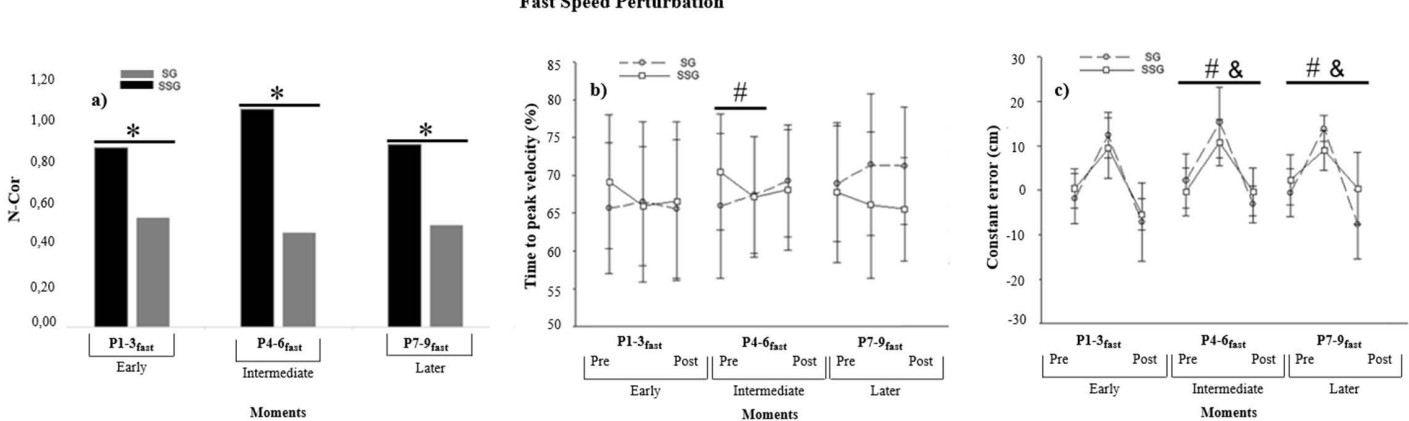

**Fig 3. Fast perturbation.** a) Mean of the Number of Corrections (N-cor). Black bars represent SSG and gray bars represent SG. b) Mean and standard deviation of the time to peak velocity (%) and c) mean and standard deviation of the constant error (cm). SG represents stabilization group practice. SSG represents superstabilization group practice. P$_{fast}$ represents perturbation with increased velocity. *Differences between groups. # SSG differences between blocks. & SG differences between blocks.

a significant interaction between groups and blocks, $F(2, 76) = 3.27$, $p = 0.04$, $\eta_p^2 = 0.07$, power $(1-\beta) = 0.60$. The post hoc detected that SSG spent less tPV (%) when P4-6$_{fast}$ was inserted than in Pre ($p < 0.05$). There was no significant difference between groups, $F(1, 38) = 0.21$, $p = 0.64$, $\eta_p^2 = 0.005$, power $(1-\beta) = 0.07$ nor blocks $F(2, 76) = 0.75$, $p = 0.01$, $\eta_p^2 = 0.72$, power $(1-\beta) = 0.17$. The analysis of CE showed a significant interaction between groups and blocks, $F(2, 76) = 4.94$, $p = 0.009$, $\eta_p^2 = 0.11$, power $(1-\beta) = 0.79$. The post hoc detected that in the intermediate moment (P4-6$_{fast}$), when the perturbation was inserted, the CE of both groups (SSG and SG) increased compared to the control trials (Pre P) ($p < 0.05$). When the perturbation was withdrawn, the CE of both groups decreased in the control trials (Post P) compared to the perturbation trials (P4-6$_{fast}$) ($p < 0.05$). More specifically, in the decomposition of interaction groups x blocks, a similar pattern of behavior between SG and SSG was seen in the Intermediate moment of the fast perturbation, but without superiority of SG. There was difference between blocks, $F(2, 76) = 86,65$, $p = 0.001$, $\eta_p^2 = 0.69$, power $(1-\beta) = 1.00$. The post hoc detected that the CE increased (i.e., later) when the perturbation was inserted (P4-6$_{fast}$) compared to Pre ($p < 0.05$). There was no significant difference between groups, $F(1, 38) = 1.29$, $p = 0.26$, $\eta_p^2 = 0.03$, power $(1-\beta) = 0.19$.

**Later Moment: Pre P x P7-9$_{fast}$ x Post P.** The analysis of the number of corrections on the third moment of P$_{fast}$ (P7-9$_{fast}$) showed that SSG performed more corrections than SG ($p = 0.02$). The analysis of tPV (%) did not show any difference between groups, $F(1, 38) = 2.99$, $p = 0.09$, $\eta_p^2 = 0.07$, power $(1-\beta) = 0.39$; blocks $F(2, 76) = 0.08$, $p = 0.91$, $\eta_p^2 = 0.002$, power $(1-\beta) = 0.06$; nor interaction, $F(2, 76) = 2.43$, $P = 0.09$, $\eta_p^2 = 0.06$, power $(1-\beta) = 0.47$. The analysis of CE showed a significant interaction between groups and blocks, $F(2, 76) = 11.99$, $p = 0.001$, $\eta_p^2 = 0.23$, power $(1-\beta) = 0.99$. The post hoc detected that in the later moment (P7-9$_{fast}$), when the perturbation was inserted, the CE of both groups (SSG and SG) increased compared to the control trials (Pre P) ($p < 0.05$). When the perturbation was withdrawn, the CE of both groups decreased in the control trials (Post P) compared to the perturbation trials (P4-6$_{fast}$) ($p < 0.05$). More specifically, in the decomposition of interaction groups x blocks, a similar pattern of behavior between SG and SSG was seen in the later moment of the fast perturbation, but without superiority of SG.

There was difference between blocks, $F(2, 76) = 67.88$, $p = 0.001$, $\eta_p^2 = 0.64$, power $(1-\beta) = 1.00$. The post hoc detected that the CE increased (i.e., later) when the perturbation was inserted (P7-9$_{fast}$) compared to Pre ($p < 0.05$). There was no significant difference between groups, $F(1, 38) = 3.27$, $p = 0.07$, $\eta_p^2 = 0.07$, power $(1-\beta) = 0.42$.

## Slow perturbation

**Early Moment: Pre P x P1-3$_{slow}$ x Post P.** The analysis of the number of corrections on the first moment of P$_{slow}$ (P1-3$_{slow}$) shown in Fig 4 indicates that SSG performed more corrections than SG ($p = 0.01$). The analysis of tPV (%) did not show difference between groups $F(1, 38) = 0.03$, $p = 0.84$, $\eta_p^2 = 0.00$, power $(1-\beta) = 0.05$; blocks, $F(2, 76) = .70$, $p = 0.49$, $\eta_p^2 = 0.01$, power $(1-\beta) = 0.16$; nor interaction, $F(2, 76) = 0.76$, $p = 0.47$, $\eta_p^2 = 0.01$, power $(1-\beta) = 0.17$. The analysis of CE showed difference between blocks, $F(2, 76) = 61.05$, $p = 0.001$, $\eta_p^2 = 0.61$, power $(1-\beta) = 1.00$. The post hoc detected that the CE increased (i.e., earlier) when the perturbation was inserted (P1-3$_{slow}$) compared to Pre ($p < 0.05$). With the withdraw of perturbation (Post), the CE decreased (closest to zero) compared to P1-3$_{slow}$ ($p < 0.05$). There was no significant difference between groups, $F(1, 38) = 0.87$, $p = 0.35$, $\eta_p^2 = 0.02$, power $(1-\beta) = 0.14$; nor interaction, $F(2, 76) = 0.20$, $p = 0.81$, $\eta_p^2 = 0.005$, power $(1-\beta) = 0.08$.

**Intermediate Moment: Pre P x P4-6$_{slow}$ x Post P.** The analysis of the number of corrections on the second moment of P$_{slow}$ (P4-6$_{slow}$) showed that SSG performed more corrections than SG ($p = 0.001$). The analysis of tPV (%) showed a significant interaction between groups and blocks, $F(2, 76) = 3.87$, $p = 0.02$, $\eta_p^2 = 0.09$, power $(1-\beta) = 0.68$. The post hoc detected that the SSG decreased tPV (%) when the perturbation was inserted (P4-6$_{slow}$) ($p < 0.05$). Moreover, the SSG spent less tPV (%) than SG on P4-6$_{slow}$ ($p < 0.05$). There was difference between blocks, $F(2, 76) = 6.15$, $p = 0.003$, $\eta_p^2 = 0.13$, power $(1-\beta) = 0.87$. The post hoc detected that the tPV (%) decreased when the perturbation was inserted (P4-6$_{slow}$) compared to Pre ($p < 0.05$). With the withdraw of perturbation (Post), the tPV (%) increased compared to P4-6$_{slow}$ ($p < 0.05$). There was no significant difference between groups, $F(1, 38) = 0.47$, $p = 0.49$, $\eta_p^2 = 0.01$, power $(1-\beta) = 0.10$. The

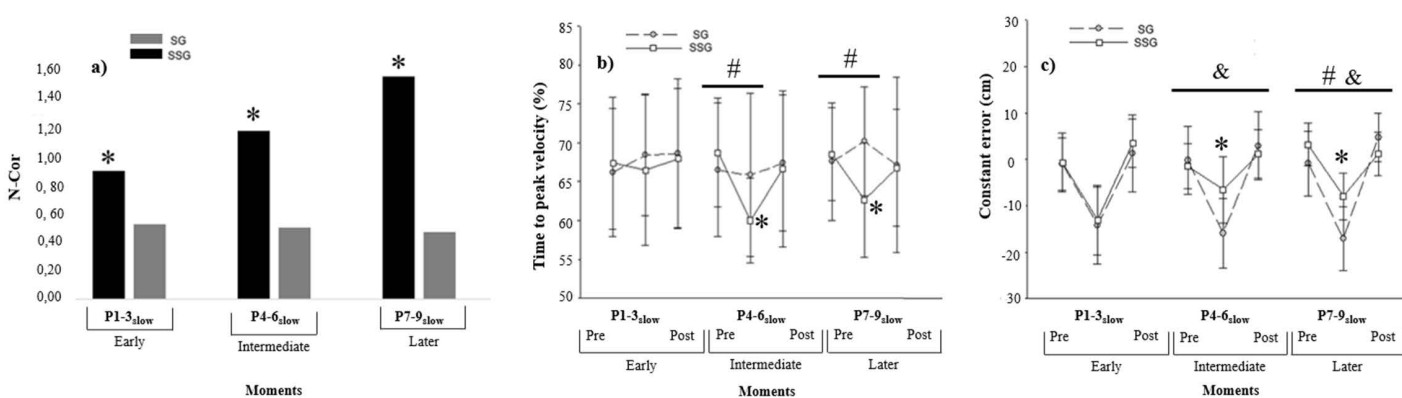

**Fig 4. Slow perturbation.** a) Mean of the Number of Corrections (N-cor). Black bars represent SSG and gray bars represent SG. b) Mean and standard deviation of the time to peak velocity (%) and c) mean and standard deviation of the constant error (cm). SG represents stabilization group practice. SSG represents superstabilization group practice. $P_{slow}$ represents perturbation with decreased velocity. *Differences between groups. # SSG differences between blocks. & SG differences between blocks.

analysis of CE showed a significant interaction between groups and blocks, $F(2, 76) = 16.89$, $p = 0.01$, $\eta_p^2 = 0.30$, power (1-β) = 0.99. The post hoc detected that the SG increased CE (i.e., earlier) when the perturbation was inserted ($P4-6_{slow}$).When the perturbation was withdrawn (Post), the CE decreasesd (closest to zero) ($p < 0.05$). Furthermore, the SSG showed lower (closest to zero) CE than SG in $P4-6_{slow}$ and similar CE in Pre and Post ($p < 0.05$). There was difference between blocks, $F(2, 76) = 48.37$, $P = 0.001$, $\eta_p^2 = 0.56$, power (1-β) = 1.00. The post hoc detected that the CE increased (i.e., later) when the perturbation was inserted ($P2_{slow}$) compared to Pre ($p < 0.05$). There was no significant difference between groups, $F(1, 38) = 3.94$, $p = 0.05$, $\eta_p^2 = 0.09$, power (1-β) = 0.49.

**Later Moment: Pre P x $P7-9_{slow}$ x Post P.** The analysis of the number of corrections on the third moment of $P_{slow}$ ($P7-9_{slow}$) showed that SSG performed more corrections than SG ($p = 0.001$). The analysis of tPV (%) showed a significant interaction between groups and blocks, $F(2, 76) = 7.16$, $p = 0.001$, $\eta_p^2 = 0.15$, power (1-β) = 0.92. SSG decreased the tPV(%) when the perturbation was inserted ($P7-9_{slow}$) ($p < 0.05$). Furthermore, the SSG spent less tPV (%) than SG on $P7-9_{slow}$ ($p < 0.05$). There was no significant difference between groups, $F(1, 38) = 1.23$, $p = 0.27$, $\eta_p^2 = 0.03$, power (1-β) = 0.19 nor blocks $F(2, 76) = 0.94$, $p = 0.39$, $\eta_p^2 = 0.02$, power (1-β) = 0.20. The analysis of CE showed showed a significant interaction between groups and blocks, $F(2, 76) = 13.29$, $p = 0.001$, $\eta_p^2 = 0.25$, power (1-β) = 0.99. SSG increased CE (i.e., earlier) when the perturbation was inserted ($P7-9_{slow}$). When the perturbation was withdrawn (Post), the CE decreasesd (closest to zero) ($p < 0.05$). Post hoc analysis revealed that the same pattern was observed for the SG when the perturbation was inserted ($P4-6_{slow}$) and when the perturbation was withdrawn (Post) ($p < 0.05$). Furthermore, the SSG showed lower CE (closest to zero) than SG on $P7-9_{slow}$ ($p < 0.05$). There was difference between blocks, $F(2, 76) = 95.22$, $P = 0.001$, $\eta_p^2 = 0.71$, power (1-β) = 1.00. The post hoc detected that the CE increased (i.e., earlier) when the perturbation was inserted ($P7-9_{slow}$) compared to Pre ($p < 0.05$) and with the withdraw of perturbation (Post), the CE decreased (closest to zero) compared to $P7-9_{slow}$. Also, there was difference between groups $F(1, 38) = 8.82$, $p = 0.005$, $\eta_p^2 = 0.18$, power (1-β) = 0.82. The post hoc detected that the SSG showed lower CE than SG ($p < 0.05$).

## Discussion

In this study, we explored the impact of two discrete levels of performance stabilization: stabilization (SG) and superstabilization (SSG), on adjustments to unpredictable perturbations in intercepting a moving target. Participants were trained

under a constant practice schedule to reach one of these stabilization levels, according to the group. Constant practice while learning a motor skill enables the formation of internal models with sufficient ability to deal with unpredictable perturbations. For example, Santos et al., (2017) [9] found that superstabilization performance provides higher competence than stabilization performance to deal with unpredictable perturbation in an isometric force control task.

After reaching the specific level of performance stabilization, participants were subjected to practice conditions featuring unpredictably timed perturbations, when the target's speed was altered randomly post-movement onset across 18 pseudorandom perturbation trials. The results showed that superstabilization notably enhanced online adjustments in response to unpredictable perturbations once superstabilization made more corrections than stabilization in the face of both fast and slow perturbations, confirming our hypothesis about the benefits of superstabilization in enhancing control mechanisms. The higher number of corrections led to superior performance in challenging conditions, specifically in facing slow perturbations.

Previous implementations of the superstabilization criterion in tasks demanding isometric force control [9] and anticipatory timing tasks prior to Exposure to both predictable [10,14] and unpredictable perturbations [3] have shown enhanced outcomes under both scenarios. This is especially significant in contexts involving unpredictable perturbations, which pose substantial challenges for reorganizing action plans [4,5]. The necessity for specialized adjustment mechanisms, such as feedback control, may clarify the benefits of superstabilization, a relationship that has not been previously explored.

Our findings indicate that superstabilization led to more frequent adjustments in scenarios involving both types of perturbations (i.e., $P_{fast}$ and $P_{slow}$), as evidenced by the acceleration analysis (Fig 2a and 3a). These adjustments entailed strategies to anticipate peak velocity closer to the movement onset, successfully implemented by the SSG in specific trials (Fig 2b and 3b). A detailed analysis of the speed and acceleration curves showed that the strategy employed by the SSG to reduce the time to peak velocity (tPV%) enabled numerous motor adjustments and resulted in significantly improved performance accuracy (i.e., lower constant error, CE) compared to the SG in $P2_{slow}$ e $P3_{slow}$ (Fig 2c and 3c). The necessity for a reduced peak velocity time in unpredictable contexts permits individuals to employ visual feedback to modulate deceleration and implement corrections [7,40,43]. Such corrections were possible only for participants in the SSG, demonstrating that achieving superstabilization enhances performance variability and adaptability, thus facilitating adjustments [10]. Moreover, superstabilization fosters a high ability to change [11], likely due to internal models that bolster the capacity to make corrections under unpredictable conditions, even though superstabilization was achieved under constant practice schedule, as in previous studies [3,9].

Effective correction in unpredictable contexts relies on adequate viewing time of the target, optimally at least 200 ms before interception [44]. If the display time is less than 200 ms, the information extracted cannot be effectively used to adjust the motor command [45], negatively impacting performance. In our experiments, participants practiced the task with movement times ranging from 200 to 250 ms. This temporal arrangement confirmed that only the achievement of superstabilization allows for the effective utilization of available time to extract relevant information, make necessary corrections, and maintain performance levels amidst perturbations.

The variances in motor control between superstabilization and stabilization are further depicted by single-subject slope curves for each level of performance stabilization. Fig 5a and 5c illustrate the velocity curves for stabilization and superstabilization, respectively, prior to Performance Increase (Pre $P_{fast}$), while Fig 5b and 5d display the adjustments during $P_{fast}$ (increase in target velocity).

Fig 6a–6d display similar velocity profiles during Phase slow ($P_{slow}$), when the target's velocity decreases. In both Phase fast ($P_{fast}$) and slow ($P_{slow}$), the SG exhibited higher velocities compared to the SSG. Unlike the SG, the SSG anticipated the peak velocity in response to the perturbation. This anticipation gave the SSG additional time to make corrections, resulting in more accurate performance.

Addressing unpredictable perturbations effectively requires rapid movement corrections within brief time intervals. One strategy to meet this challenge involves employing a hybrid model of motor control. In this model, a basic motor command

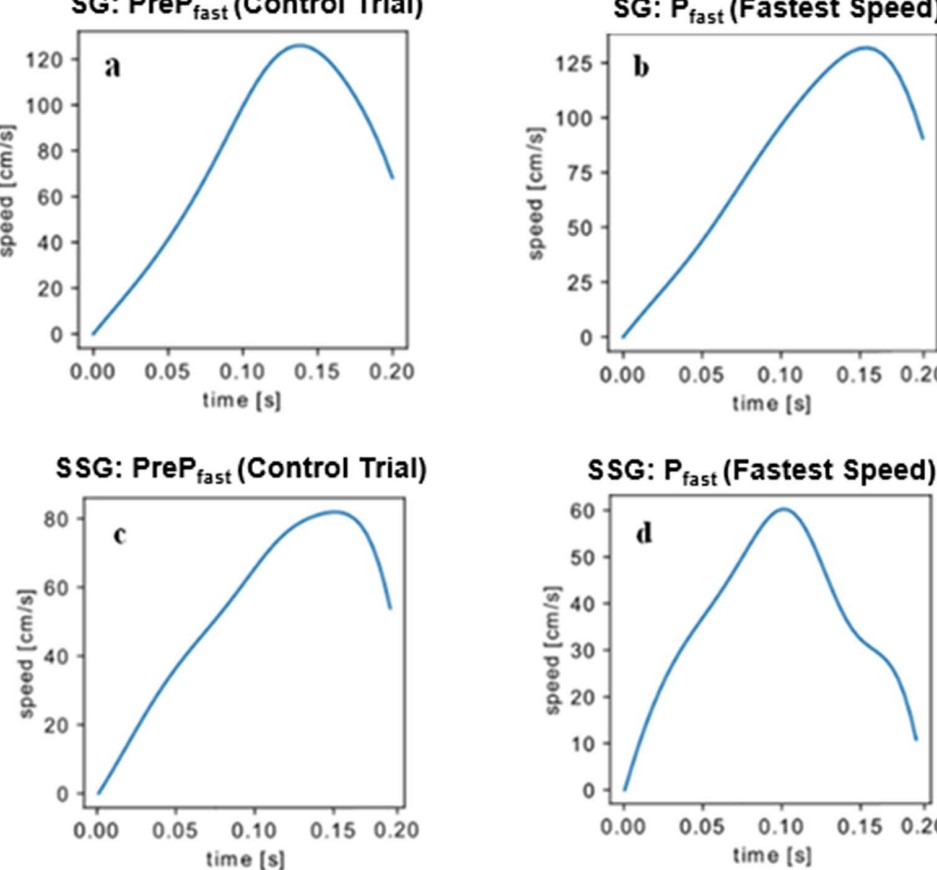

**Fig 5. Velocity of movement (cm.s$^{-1}$).** a. Represents the behavior of the SG during a control trial. b. Represents the behavior of the SG during a trial with $P_{fast}$ (fast velocity). c. Represents the behavior of the SSG during a control trial. d. Represents the behavior of the SSG during a trial with $P_{fast}$ (fast velocity).

is pre-programmed before movement initiation. Following this, the motor command is continually monitored and adjusted based on visual input, refined by a forward model embedded within internal models for optimal kinematic and dynamic control [46,47]. This experiment suggests that superstabilization notably enhances the ability to extract visual information critical for responding to unpredictable perturbations. The visual system collaborates with internal models to formulate predictions [48]. Specifically, the peripheral retina and the fovea detect cues and generate error signals for rapid corrections during ongoing movement [49], as observed in our target experiment. These cues and error signals bolster the forward model, enhancing its capacity to manage the internal feedback circuit and potentially diminishing delays associated with sensory feedback in subsequent attempts [46,48]. Such mechanisms likely account for the SSG's superior handling of unpredictable perturbations, warranting further investigation within shorter time frames than those manipulated in the current study.

Our analysis revealed that both groups executed online corrections. However, the SSG made more adjustments than the SG, as indicated by Figs 2c and 3c data. While the SG did respond to perturbations, this group seems to have relied predominantly on information from previous attempts to control the task (i.e., pre-perturbation trials), suggesting a less effective adjustment strategy. This reliance and influence of previous trials on the motor control of current trials are not uncommon in interceptive tasks, such as simulated baseball batting [30] and other tasks, lsuch as in reaching task with

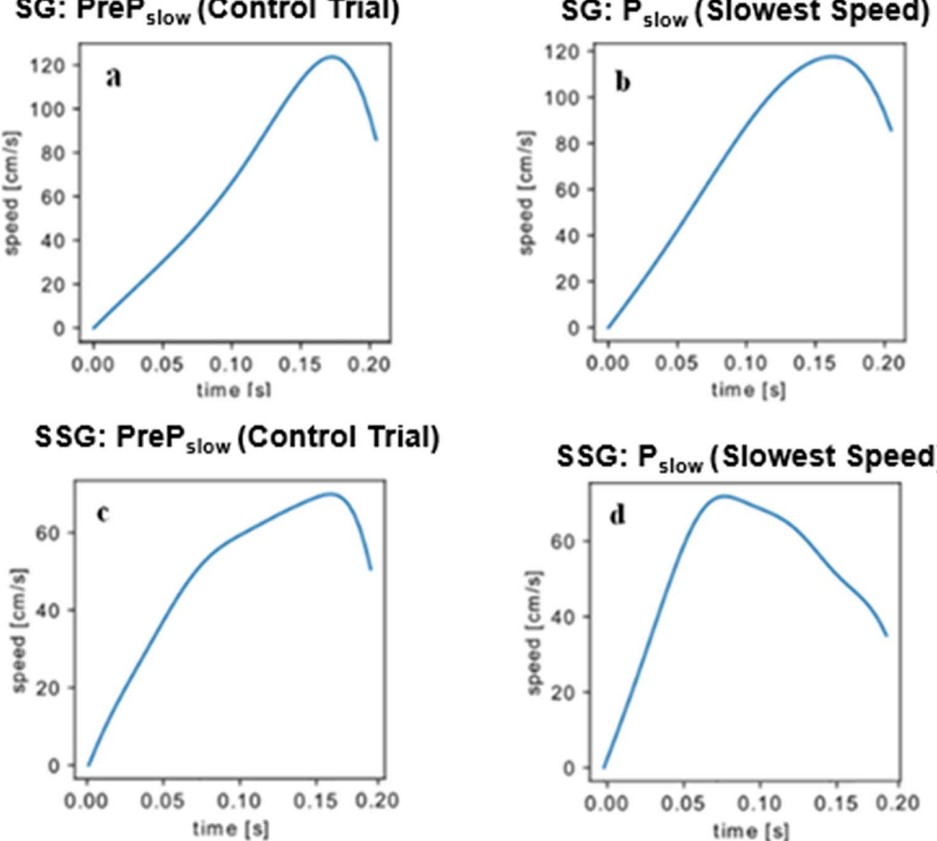

**Fig 6. Velocity curves (cm.s⁻¹).** a. Represents the behavior of the SG during a control trial. b. Represents the behavior of the SG during a trial with P slow ($P_{slow}$). c. Represents the behavior of the SSG during a control trial. d. Represents the behavior of the SSG during a trial with $P_{slow}$ (slow velocity).

unpredictable perturbation in a curl force fields [6]. However, it may lead to errors in unpredictable contexts, necessitating rapid post-detection corrections. Consequently, the SG demonstrated inefficiencies in the number of corrections needed to adjust actions, highlighting the SSG's superiority.

Our findings suggest that achieving a higher level of stabilization provides the sensorimotor system with more effective strategies for managing unpredictable perturbations, as evidenced by both kinematic and performance data. Moreover, these results indicate that extending practice beyond stabilization fosters internal models related of the task, capable of leveraging real-time environmental information rather than solely depending on past experiences. This insight underscores how elevated skill levels can assist performers in navigating complex motor challenges. Future investigations should further explore the impact of stabilization levels, as this knowledge has significant theoretical and practical implications across various motor learning domains, such as sports and rehabilitation, where adaptable motor behaviors are crucial.

## Limitations and future perspectives

We recognize that our study has limitations. The method by which we introduced perturbations enabled us to explore the effects of different performance stabilization levels in response to unpredictable perturbations, but it is limited to two magnitudes. However, limiting to two magnitudes restricts the possibilities of extrapolating our results in different situations,

since unpredictable perturbations of varying magnitudes are commonplace in everyday life and in sports contexts. Consequently, incorporating perturbations that account for predictability and diverse magnitudes could extend our findings, specially when considering using our findings as a starting point for decision-making in pratical contexts involving interception tasks for moving targets, which is very common in sports.

Furthermore, beyond levels of performance stabilization, we suggest exploring variations in practice schedules, such as varied practice. Research indicates that practice schedules, beyond merely the quantity of practice, can significantly affect the ability of internal models [50]. This flexibility is essential for adapting effectively to perturbations, suggesting that alterations in training regimens could further elucidate the dynamics of motor control and performance adaptation. This research is being conducted in our laboratory.

## Conclusion

In line with our hypothesis, our findings demonstrate that achieving superstabilization, not merely stabilization, enhances adjustments to unpredictable fast and slow perturbations and performance outcomes, specifically in face to slow perturbation. Superstabilization appears to cultivate more robust and changeable internal models of the task, enabling the sensorimotor system to extract and utilize online information more effectively and make rapid movement adjustments within short intervals.

Despite our progress in understanding motor control in tasks involving moving target interception under conditions of unpredictability and limited time for adjustments, we recommend that future studies explore manipulating the interval following the introduction of a perturbation. Reducing the time between the onset of the perturbation and its interception could challenge the ability to make adjustments using external feedback, thus necessitating reliance solely on the internal feedback mechanisms of the internal model. Such manipulation could shed light on the internal model's capabilities when practice extends beyond basic performance stabilization.

## Supporting information

**S1 Data. Individual data of the volunteers of both groups, performance stabilization (SG) and performance superstabilization (SSG), in the variables number of corrections (NCor) during perturbations, relative time to peak velocity (tPV%) and constant error (CE) in pre-perturbation, perturbation and post-perturbation.**
(XLSX)

## Author contributions

**Conceptualization:** Crislaine Rangel Couto, Cláudio Manoel Ferreira Leite, Carlos Eduardo Campos, Leonardo Luiz Portes, Herbert Ugrinowitsch.

**Data curation:** Cláudio Manoel Ferreira Leite, Carlos Eduardo Campos, Cíntia de Oliveira Matos, Suziane Peixoto Santos.

**Formal analysis:** Crislaine Rangel Couto, Cláudio Manoel Ferreira Leite, Carlos Eduardo Campos.

**Investigation:** Crislaine Rangel Couto.

**Methodology:** Crislaine Rangel Couto, Herbert Ugrinowitsch.

**Project administration:** Hani Camille Yehia, Herbert Ugrinowitsch.

**Resources:** Herbert Ugrinowitsch.

**Software:** Crislaine Rangel Couto, Leonardo Luiz Portes.

**Supervision:** Herbert Ugrinowitsch.

**Validation:** Herbert Ugrinowitsch.

**Writing – original draft:** Crislaine Rangel Couto, Cláudio Manoel Ferreira Leite, Carlos Eduardo Campos, Cíntia de Oliveira Matos, Suziane Peixoto Santos, Natália Fontes Alves Ambrósio.

**Writing – review & editing:** Leonardo Luiz Portes, Hani Camille Yehia, Herbert Ugrinowitsch.

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
