## [Decision Letter · Decision Letter 0]

8 Jan 2025

Dear Dr. Ugrinowitsch,

Thank you for submitting your manuscript to PLOS ONE. After careful consideration, we feel that it has merit but does not fully meet PLOS ONE’s publication criteria as it currently stands. Therefore, we invite you to submit a revised version of the manuscript that addresses the points raised during the review process.

We look forward to receiving your revised manuscript.

Kind regards,

Job Fransen

Academic Editor

PLOS ONE

Journal Requirements:

2. Please include captions for your Supporting Information files at the end of your manuscript, and update any in-text citations to match accordingly. Please see our Supporting Information guidelines for more information: http://journals.plos.org/plosone/s/supporting-information .

Reviewers' comments:

Reviewer's Responses to Questions

**Comments to the Author**

1. Is the manuscript technically sound, and do the data support the conclusions?

Reviewer #1: No

Reviewer #2: Yes

2. Has the statistical analysis been performed appropriately and rigorously?

Reviewer #1: Yes

Reviewer #2: Yes

3. Have the authors made all data underlying the findings in their manuscript fully available?

Reviewer #1: Yes

Reviewer #2: Yes

4. Is the manuscript presented in an intelligible fashion and written in standard English?

Reviewer #1: Yes

Reviewer #2: Yes

Reviewer #1: PONE-D-24-43982

Practice beyond performance stabilization increases the use of online adjustments to unpredictable perturbations in an interceptive task

Couto, C., Leite, C., Campos, C., Portes, L., Matos, C., Santos, S., Ambrósio, N., Yehia, H. C., & Ugrinowitsch, H.

This study focuses on the effects of performance stabilization on motor adjustments to unpredictable target velocity during an interceptive task. Particularly, authors separated two groups based on their respective achievement of performance criteria (groups of stabilization and superstabilization). The manuscript is interesting and shows some novelty in bringing to debate the effects of level of participants’ skill, learning, or performance during an interceptive action, which certainly will attract attention of the readers of PLOS One. Overall, the manuscript is well-written, organized, based on appropriate data analysis and interpretation. However, there is a variety of weakness points and respective concerns that I have, along with some suggestions of improvement of the current version of the manuscript, as described below (following the order of appearance in the text):

i. As pointed out above, the study focuses on the stabilization process as a consequence of motor practice. But the characterization of the level of stabilization of both groups is not clearly shown. How did performance changed (or improved) from the achievement of intercepting the target three trials in a row to the achievement of intercepting the target three trials in a row six times (respectively, for stabilization and superstabilization groups)? Comparing the groups, how was variability reduced to allow performance to be considered superstable? Authors do not report performance scores for each group other than the variables number of corrections, time to peak velocity, and constant error with respect to Perturbation I (Fig. 3) and Perturbation II (Fig. 4). Interestingly, the values of constant error of both groups for PrePI1 (Fig. 3c) and PrePII1 (Fig. 4c), which are the first information available since the separation of groups, do seem very similar. It is necessary to add explicit data on the differences of the interceptive task performance between the two groups; also, providing details on the amount of practice involved in the preparation phase for both groups is necessary. Additionally, the criterion “intercepting the target three trials in a row six times” is not sufficiently clear. Provide details on what exactly “six times” mean in terms of the circumstances of the proposed method; it does not seem to be 18 times in a row (lines 194-195). Also, there is an additional definition of criteria for separating the groups (see lines 197-201), stating that an error smaller than or equal 5 cm in 200 trials for the stabilization group and an error smaller than or equal 5 cm in 320 trials for the superstabilization group. How both criteria for separating the groups were simultaneously applied? A third criterion was also used (movement time between 200-250 ms); although the use of feedback is described for different time ranges, what would happen to trials out of the expected duration (200-250 ms)? Overall, this description is very confusing. Please, clarify this issue.

ii. In a similar vein as above, there is a central question regarding the proposed design. If the main interest of authors was to understand the effects of practice on the performance of an interceptive task when target velocity changes throughout its arrival time, why did not participants practice (during preparation or acquisition phase of the study) in the same conditions (with change in target velocity) instead of practicing under target constant velocity? So the criteria for defining both groups would be specific for changes in target velocity, the proposed goal of the study. Authors need to justify the adequacy of such a design and the consequences of their choices; authors did not mention this aspect in the Introduction and/or in the Discussion section.

iii. Lines 62-66 (Introduction): “Predictable perturbations are identified before the movement onset; consequently, the action planning may contain the changes necessary to adapt to the perturbation. Conversely, unpredictable perturbations are identified only after the onset of movement; consequently, the action planning has to be reorganized after its triggering to adapt to the perturbation”. I believe that there is a misunderstanding on the distinction between motor action and visual stimuli. Stimuli may be predictable or unpredictable while stimuli perception may occur prior or after movement onset; time constraints involved in motor action and the time stimuli are available to the perceiver may vary. The suggested links between predictable – before onset and unpredictable – after onset do not seem appropriate. Please, consider to rephrase these sentences, making the concepts adequate.

iv. Lines 245-253 (Material and Methods): there are various aspects of the exposition phase that are not clear and require some discussion as follows: a first aspect refers to nomenclature. Instead of using numbers to differentiate between increase or decrease of target velocity (authors used PI and PII, respectively) and among early, intermediate, or late exposition (authors used, e.g., PI1, PI2, PI3), the terms themselves could have been used to facilitate understanding. A second aspect refers to the reason of separating data as level of the design. For example, why were three levels of responses for increase/decrease in velocity (early, intermediate, and late) used? The statistical analysis (ANOVA) proposed does not use these three levels as an independent variable of the design; why did these data were collected? Presumably, these three levels were averaged to be entered to the ANOVA. In short, it is not clear what is the importance of early, intermediate, and late exposure in this study. In the same manner, it is not clear why pre/post blocks (both with constant target velocity) were used in the experimental design; although these levels were used as independent variable, there is no explanation for using a block of trials with change in target velocity “isolated” between two (pre and post) blocks of trials with constant target velocity. The rationale of the adopted design needs to be clearly explained to readers.

v. Lines 379-380 (Discussion): “… on adjustments to unpredictable perturbations”. The amount of practice provided by the protocol was in trials with target constant velocity (predictable situation). Performance stabilization criteria for each group was reached in those predictable trials. It is not clear, and should be emphasized and discussed by authors, that stabilization criteria were achieved with practice under predictable situation.

vi. Line 384 (Discussion): “18 random trials”. These trials were not completely random because there were blocks of 3 trials to accommodate pre and post unpredictable block of trials. Provide details of the experimental design.

vii. Lines 386-388 (Discussion): “Thus confirming our hypothesis about the benefits of superstabilization in enhancing control mechanisms and performance outcomes”. This conclusion can be challenged. The adjustments were in response to unpredictable and predictable (pre and post) target velocities. In fact, the predictable ones represent 2/3 of these data submitted to ANOVA. The superstabilization also affected the predictable trials (main effect of group). In the current experimental design, benefits of superstabilization should be indicated by significant group by blocks interaction. Overall, considering three dependent variables (number of corrections, tPV, CE) and three exposures, nine significant group by blocks interactions could be found. However, for the Perturbation I, only three significant interactions were found; similarly, for the Perturbation II, only four significant interactions were found. Thus, this does not confirm the hypothesis as stated by authors. These aspects should be included in the Discussion section.

viii. Lines 478-485 (Limitations and Future Perspectives): The limitation is not clearly described. Review text.

ix. Line 505 (Conclusion): No previous reference to “forward models”. Introduction and Discussion sections could provide some background on this, if this reference is really needed.

Minor points:

lines 74-75: “… consistent internal representation of the limbs involved in the task and environmental dynamics…”. The representation should refer to limb movements’ program and/or parameters and not to the limb itself.

line 107: “the successful interception of moving interception…”. Repeated words.

lines 113-114: “…the onset of the onset…”. Repeated words.

line 123: “…relation between the level of performance stabilization level”. Repeated words.

Line 384: “The results demonstrated that…”. Demonstration seems more a result of logical reasoning, while “show” can be used as direct link to the data themselves. “The results showed that…” can be used instead.

Line 280 (as example): the term “main interaction effects” is frequently used in the Results section and is not appropriate. Traditionally, ANOVA has “main effects” (groups and blocks, in this study) and “interaction” (group by block, in this study).

Reviewer #2: General comments:

The present study examines the impact of learning or practice level on corrections to interceptive movements following an unexpected shift in target velocity. Broadly speaking, the authors highlight more rapid (%tPV) and frequent corrections (N-corr) following a state of “superstabilization” compared to the lesser “stabilization”. This study nicely highlights the mechanisms or underpinning processes that are related to corrections in interceptive tasks. The authors adopt a well-controlled study design using contemporary techniques. It is very well written. That said, there are numerous points identified that require further attention before a more substantative review can take place and an acceptance recommendation can be made. Please see below for further details.

Major comments:

Introduction:

1) What is “performance stabilization” or “superstabilization”? It should be more clearly stated or defined from the outset. This is particularly relevant when consider the seemingly menial differences between the performance levels of these learning categories (i.e., 3 right trials for stabilization and 6 right trials for superstabilization; Ln. 192-195).

2) There should be mention of the plentiful work done around dealing with unexpected online perturbations in near-aiming tasks (with set targets) [Elliott et al., 2018, Crevecoeur, Scott & Cluff, 2019].

3) While forming predictions surrounding “increased number of phase switches in the acceleration curve and a reduced time to reach peak velocity” (Ln. 132-136), it remains unclear why this may be the case due to the absence of any prior explanation or rationale.

Method:

4) As I understand it, there were a total of 129 trials with only 18 perturbation trials (x9 trials each positively [PI] and negatively [PII] accelerating) (Ln. 218-221). Is this really enough or adequately representative of a corrective response?

5) Why limit the study or analysis of “exposition” to CE and tPV%? Why not N-corr?

6) If “exposition” is of interest, then why not incorporate it as a factor in a more complex omnibus ANOVA (e.g., group x exposition x trial). As it stands, the inferential stats are somewhat fragmented and potentially limit any over-arching conclusions that could be made.

Results:

7) Interactions between group and block are identified (Ln. 289 and Ln. 303), although the pairwise comparisons reported in the text does do not shed any further light on the direction of these interactions. Along these lines, some of pairwise comparisons that are drawn appear inconsistent with some making comparisons within- (Ln. 290-291) and others between-measures (Ln. 336-337).

8) Along these lines, the symbol (*) indicating statistically significant pairwise differences in both Figures 3 and 4 look like they could be highlighting differences between-measures (i.e., SSG vs. SG), although the figure captions allude to mere “interactions differences”. If comparisons are drawn within- and between-measures, then use different symbols (e.g., [*] between, [†] within) in different locations (e.g., adjacent to data points for between, and adjacent to lines for within) accordingly.

9) In the multiple reports for CE, the direction of effects can sometimes conflate increases/decreases with error away from 0 (i.e., perfection). Rather than being too literal in describing the direction on the scale, perhaps it would be more informative to indicate the implications on error (e.g., “more negative/positive error”, “more susceptible to the negatively accelerating target”). This is not to say that direction is not important to observe because it indicates the impact of the perturbation (e.g., positively accelerating target should delay the response and add (+) to the CE score), although the nature of responses in terms of distance from 0 should be highlighted more often.

Minor comments:

Method:

1) Why an effect size of 0.3 (Ln. 145)? Based on what prior effects and related studies?

2) Does “reability” mean to be “reliability” (Ln. 146)??

3) Given the disparity between the aiming movement surface and projector display (Ln. 156-160), what was the movement gain or cursor-to-pen mapping?

4) The mention of “target velocity” and “time window” is somewhat lost on me (Ln. 186 and Ln. 222-223). What is the “time window” in reference to?

5) I take it that a 145 cm/s target velocity with a 210 cm amplitude (until reaching the intercept zone) would mean the target moves for over 1 sec (1.45 secs to be precise). Therein, with a 200-250 ms MT, then participants should have initiated their movements some time way after target motion initiation. Please clarify.

6) With a 200-250 ms MT and 27.7 cm movement amplitude, then the intended limb velocity was around 138 cm/s. Please clarify.

7) Figure 2 – Having a “blue target” and “red effector” seems odd to me, particularly when they were coloured yellow and green, respectively (like in Figure 1).

8) There does not seem to be any prior mention of “exposition” (Ln. 245)? Does it need to be “exposure”?

9) As I understand it, there were x9 trials each positively (PI) and negatively (PII) accelerating trials (Ln. 219-220). With 3 “blocks” of “exposition” (Ln. 251-253), then would that mean there were x3 trials devoted to each level of block (e.g., Pre PI2 = x3 trials)?

References:

Crevecoeur, F., Scott, S. H., Cluff, T. (2019). Robust control in human reaching movements: a model-free strategy to compensate for unpredictable disturbances. Journal of Neurophysiology, 39(41), 8135-8148.

Elliott et al. (2017). The multiple process model of goal-directed reaching revisited. Neuroscience & Biobehavioral Reviews, 72, 95-110.

**Do you want your identity to be public for this peer review?** For information about this choice, including consent withdrawal, please see our Privacy Policy

Reviewer #1: **Yes: ** SÉRGIO TOSI RODRIGUES

Reviewer #2: No

---

## [Author Response · Author response to Decision Letter 1]

7 Mar 2025

We are grateful to the reviewers for their suggestions and the opportunity to improve the manuscript to make it clearer in this version. The comments made us think and write a lot about them, so some answers are quite long, and the whole file is nearly a new manuscript. Sorry about that. However, the manuscript improved significantly in our analysis. Thank you again about your comments. All changes in the manuscript are shown in blue.

REVIEWER 1

Major comments:

The first comment involves many concepts and method. So, to provide a clearer response, we have divided comment 1 into subcomments: 1a, 1b, 1c, 1d, 1e, 1f, and 1g.

Comment 1a) As pointed out above, the study focuses on the stabilization process as a consequence of motor practice. But the characterization of the level of stabilization of both groups is not clearly shown. How did performance changed (or improved) from the achievement of intercepting the target three trials in a row to the achievement of intercepting the target three trials in a row six times (respectively, for stabilization and superstabilization groups)?

Answer: This is our study's central point and we appreciate your comment. We have now worked on clarifying it (lines 70-79). Performance is considered stable when errors remain within a predefined range of accuracy defined as acceptable

for a given task, and it is task-dependent. In our study, after conducting many pilot studies, we identified that volunteers were unlikely to repeat a performance with an error smaller than -5 cm to +5 cm from the center of the target more than three trials in a row over 500 attempts. Therefore, we adopted the criterion for this task as three consecutive trials with an error between -5 cm and +5 cm. This same criterion has been used in previous studies (Couto et al., 2021; Campos et al., 2022) for this task.

Specifically considering the two levels of stabilization, for performance to be stable (i.e., consistently reproduced within a small margin of error), there must be a spatiotemporal standardization of the movement, resulting in correct responses in consecutive trials, which allows us to consider that performance stabilization has been achieved (Burdet et al., 2006; Corrêa et al., 2015; Loschiavo-Alvares et al. 2023). This rationale has been applied in studies involving complex coincident-timing tasks, in which performance stabilization is reached when the error is equal to or less than 30 ms, and this level of performance is repeated for three consecutive trials (Fonseca et al., 2012; Ugrinowitsh et al., 2011; Campos et al., 2022; Loschiavo-Alvares et al., 2023). In an isometric force control task, we also ran pilot studies and found that participant could perform no more than four consecutive trials with an error smaller than 5% of the task goal (i.e., reach 40% of the maximum force). So, we adopted this performance criterion in a force control task (Santos et al., 2017). As described above, all these criteria were established following pilot studies to identify how volunteers perform according to the task.

Considering superstabilization requires another criterion, but pilot studies have shown that due to task constraints, performance accuracy does not increase indefinitely, as this would require a level of expertise. We would never be able to perform this study if we consider expertise as 10 years of practice or 10 thousand trials (Ericsson & Smith, 1994). Therefore, superstabilization was operationally defined as repeatedly maintaining stabilized performance. Again, based on pilot studies, we have adopted repeating the stabilization criteria for six blocks, as adopted in previous studies (Fonseca et al., 2012; Ugrinowitsch et al., 2011; Corrêa et al., 2015; Couto et al., 2021; Campos et al., 2022).

Previous studies (Ugrinowitsch et al., 2014 and Couto et al., 2021) have shown that, despite statistically equal accuracy during the learning phase, greater performance variability is identified when reaching superstabilization compared to those who practiced until stabilization. Still, most other studies that compared performance in stabilization with superstabilization did not find this behavior. This may happen because the differences can be found in control measures not in performance, and it depend on the characteristics of the task as well. However, this discussion about what happens in the phase prior to the insertion of perturbations was not the aim of the present study but rather to investigate behavior in the face of unpredictable perturbations.

Given the established criterion, we did not expect to find differences in performance during the first phase of the study (i.e., learning phase) but rather when participants had to face perturbations. Indeed, studies have shown that under such conditions, groups that practiced until reaching the superstabilization performance criterion (i.e., six blocks of three or four consecutive successful trials) could better deal with the perturbations than those that practiced only until reaching the performance criterion for what we named stabilization (Fonseca et al., 2012; Ugrinowitsch et al., 2014; Corrêa et al., 2015; Santos et al., 2017).

Comment 1b) Comparing the groups, how was variability reduced to allow performance to be considered super stable?

Answer: Stability can be interpreted in different ways. In the present study, we are not interpreting stability only as associated with reduced variability, but also concerning increased accuracy. More specifically, the definition of different stabilization levels is based on absolute spatial error within a quite small error range (accuracy), range repeated a few times (consistency). A previous study conducted by our group (Couto et al., 2021) with this same task shows that both the variability of performance and control measures, of the levels of superstabilization and stabilization, decreased throughout the learning phase, even with the superstabilization group showing greater variability of performance than the stabilization group. Here is important to highlight that the criterion adopted to indicate stability is highly stringent. It often makes it impossible to identify a difference between stabilization and superstabilization during the learning phase. Once more, maybe we could find differences between stabilization level and expertise, but it is a question for another study.

Comment: 1c) Authors do not report performance scores for each group other than the variables number of corrections, time to peak velocity, and constant error with respect to Perturbation I (Fig. 3) and Perturbation II (Fig. 4). Interestingly, the values of constant error of both groups for PrePI1 (Fig. 3c) and PrePII1 (Fig. 4c), which are the first information available since the separation of groups, do seem very similar. It is necessary to add explicit data on the differences of the interceptive task performance between the two groups.

Answer: As we understand it, this comment refers to the differences resulting from the manipulation of the two levels of stabilization, so we will proceed accordingly.

Considering that both stabilization levels have the same performance criterion to stop practicing, both are expected to have similar performance, and the difference in task performance will only appear when the perturbation is introduced and/or after its removal. Before the perturbation, the only distinction between groups that achieve stabilization and superstabilization is in performance variability, as demonstrated in some previous studies (Ugrinowitsch et al., 2014; Couto et al., 2021).

Furthermore, the expected explicit difference between two levels of stabilization in the first phase could appear when comparing an expert with someone who has merely stabilized the performance, which is not the case in the present study. As previously mentioned, the stabilization levels may not have a difference in performance accuracy in the learning phase since the practice ends when participants reach the criterion established for the task, which is the same for both groups (i.e., three trials in a row with AE < + 5cm). Interestingly, the differences appear when they need to deal with perturbations. Probably this difference is because by prolonging the practice beyond stabilization (i.e., superstabilization), when participants have more opportunities to obtain information about the target's displacement speed and the interception location, and thus improve their skill in the task, observed when facing perturbations. For example, when the perturbation was inserted, Fonseca et al. (2012) and Ugrinowitsch et al. (2014) found a difference in the organization of the components of the complex task, and Santos et al. (2017) found a difference in co-contraction, although none of the studies showed difference between the two levels of stabilization during the learning phase. It means that superstabilization allowed for more information about the task and its execution, facilitating adjustments in the control mechanisms and achieving the same performance. These results show the importance of continuing to investigate the difference in stabilization levels that allow for better performance or better control when facing perturbations.

In this study, we adopted performance and control mechanism measures. Like some previous studies, the PRE Pfast (in the first version named PI) data did not show any difference between the stabilization and superstabilization levels in the

First phase (we explained it above). For example, the constant error increased when facing the perturbation and return to the level of Pre-Perturbation in the control trial – PosP (speed used in the learning phase).

Considering the no differences between groups during PreP, one explanation is the performance criterion adopted to finish the first phase: the error should be < + 5 cm. As we explained before, this error criterion was adopted based on pilot studies, and it seems correct because the difference between stabilization and superstabilization can only be detected under unpredictable conditions. That is the reason we investigated this question in the manuscript. If we had chosen another error criterion to stop practicing, e.g., + 20 cm, there would probably be differences in PreP that would make it unfeasible to investigate the effects of levels of stabilization during unpredictable perturbations.

1d) Also, providing details on the amount of practice involved in the preparation phase for both groups are necessary.

Answer: The focus of this study is on the second phase and not on the first. Previous studies (e.g., Couto et al., 2021 and Campos et al., 2022) have already demonstrated that achieving stabilization and superstabilization of performance in the same task and under the same practice conditions as our study ensures the achievement of two different levels of sensorimotor system functionality — that is, two different levels of performance stabilization. This serves as the starting point of the present study, and not the process of reaching the different levels.

This is an important comment because it differentiates our study from many others related to motor learning. Most of the studies adopt a specific number of trials during the learning phase (McGarity-Shipley et al., 2023; Cruz et al., 2024). This procedure gives good control over the amount of practice but not the level of learning. It means that some participants can have different levels of learning in the task, which can influence the results. Differently, some studies use a performance criterion in the learning phase (Santos et al., 2017; Campos et al., 2022). This procedure gives good control over the level of learning because all participants stop practicing at a similar level of performance. Consequently, it gives better control over the effects caused by the variable tested (e.g., feedback, attentional focus) because any difference observed after practicing is caused by the independent variable and not from differences in the level of learning in the task.

Apart from that, we organize data related to the number of trials to reach the criterion of performance for both levels of stabilization. The figure below illustrates the average number of trials achieved by both groups (gray bar: SG and black bar: SSG). The test t revealed that the SSG performed more trials than SG t(38) = -3.967, p=0.0003. However, we emphasize once again that the focus of this study is on the second phase, when perturbations were inserted, and not on the first.

Comment: 1e) Additionally, the criterion “intercepting the target three trials in a row six times” is not sufficiently clear. Provide details on what exactly “six times” mean in terms of the circumstances of the proposed method; it does not seem to be 18 times in a row (lines 194-195_Now lines201-202).

Answer: Perfect. We do agree with the comment, and have modified the writing on line 202 to make it clearer for the reader. In this new version, you will find: “The Superstabilization Group practiced until they reached the same performance criterion (three right trials in a row), which should be repeated six times throughout the phase. So, Stabilization should perform one block of three right trials in a row, and Superstabilization should perform the same criterion (i.e., three correct trials in a row) for six times”

Comment: 1f) Also, there is an additional definition of criteria for separating the groups (see lines 197-201_Now lines 204-208), stating that an error smaller than or equal 5 cm in 200 trials for the stabilization group and an error smaller than or equal 5 cm in 320 trials for the superstabilization group. How both criteria for separating the groups were simultaneously applied?

Answer: In fact, the criteria was not applied simultaneously. In lines 195-197 cite those participants were randomly assigned into two groups, SG and SSG. Lines 199-203 report the criterion for each group and the number of trials allowed to reach the criterion. SG had 200 trials because this group should perform the criteria only once. The SSG had 320 trials because it should repeat the criteria six times. In fact, participants spent more time learning the VIT and reaching the performance criteria. That is the reason of 200 trials. The SSG should repeat the performance criteria for six times, it was not necessary to improve the amount of practice by six. All these numbers were identified in pilot studies. Participants that did not reach the performance criteria were excluded from the experiment, as happened with two participants (description between lines 216 and 217).

We have revised the section (lines 208–213) to facilitate the reader’s understanding.

Comment: 1g) A third criterion was also used (movement time between 200-250 ms); although the use of feedback is described for different time ranges, what would happen to trials out of the expected duration (200-250 ms)? Overall, this description is very confusing. Please, clarify this issue.

Thank you for the comment. We have revised it (lines 218–223) for clarity. Below there is the explanation concerning the movement time control.

In this study, the movement time does not refer to an independent variable but rather a control variable. We aimed to identify the strategies used and, from them, infer the control mechanism when groups that achieved two different levels of performance stabilization were exposed to perturbations requiring modifications of the prior planning after movement onset.

In interception tasks involving moving targets, movements lasting 150 milliseconds are considered fast (Tresilian, 1995). However, restricting movement time to 150 milliseconds would not allow us to identify control strategies that rely on sensory feedback to adjust planning before perturbations and, consequently, the use of the two possible mechanisms of control. Therefore, to determine whether participants predominantly used pre-programmed control or online feedback when facing perturbations inserted immediately after movement onset — within a short time window for corrections—they were required to learn a movement time ranging between 200 and 250 milliseconds.

Once more, the procedures were determined through pi

---

## [Decision Letter · Decision Letter 1]

11 Apr 2025

Dear Dr. Ugrinowitsch,

Thank you for submitting your manuscript to PLOS ONE. After careful consideration, we feel that it has merit but does not fully meet PLOS ONE’s publication criteria as it currently stands. Therefore, we invite you to submit a revised version of the manuscript that addresses the points raised during the review process.

We look forward to receiving your revised manuscript.

Kind regards,

Job Fransen

Academic Editor

PLOS ONE

Additional Editor Comments:

Dear authors

Reviewer two has raised additional concerns which I would need to see addressed properly before recommending acceptance of your manuscript. Please address those at your earliest convenience.

Reviewers' comments:

Reviewer's Responses to Questions

**Comments to the Author**

Reviewer #1: All comments have been addressed

Reviewer #2: (No Response)

2. Is the manuscript technically sound, and do the data support the conclusions?

Reviewer #1: Yes

Reviewer #2: Yes

3. Has the statistical analysis been performed appropriately and rigorously?

Reviewer #1: Yes

Reviewer #2: Yes

4. Have the authors made all data underlying the findings in their manuscript fully available?

Reviewer #1: Yes

Reviewer #2: No

5. Is the manuscript presented in an intelligible fashion and written in standard English?

Reviewer #1: Yes

Reviewer #2: Yes

Reviewer #1: Authors have fully answered the majority of points raised and provided an improved version of the manuscript. I appreciated the detailed explanations authors have made. Although I do not completely agree with part of the reasoning/answers presented by authors, it is clear that this new version of the manuscript is easier to read and flows better. I understand that the most relevant aspects which needed change were sufficiently improved to publication standards.

Reviewer #2: General comments:

The authors have made robust attempts to handle any comments. However, there mostly existing issues that I believe remain outstanding. Perhaps most obvious are the comments related to analysis choices and the outline of key interactions within the text. I have further detailed below:

Major comments:

Introduction:

1) While stabilization has been operationally defined within the manuscript, I find the explanation offered in the response letter to be more informative. Perhaps include some of this content, including details of empirical findings (e.g., Campos et al., 2022), in order to better envisage the state of (super-)stabilization.

Method:

2) I remain unconvinced by the failure to incorporate “exposure” into the analysis. It has been explained within the response letter that changes/learning were not of primarily concern when it came to the perturbation trials, although it begs the question: why consider “exposure” in the first place including the separation of perturbation trial blocks (i.e., x3 trials per block)?

Results:

3) While the direction of effects has been made much clearer, the failure to decompose interactions within the text still remains. Instead, I’m having to solely observe the between-group differences purported in Figures 3 and 4 (via symbols) because the text mostly describes common differences between blocks within SG and SSG groups (synonymous with a block main effect) (Ln. 312-315, 327-330). When it comes to the interaction, I would advise focusing the report on the differences highlighted within the fore mentioned figures (e.g., CE in fast speed perturbation: SSG < SG for P, SSG > SG for Post).

Minor comments:

Introduction:

4) I may have misquoted the year of citation for Elliott et al. Here is the reference:

Elliott et al. (2017). The multiple process model of goal-directed reaching revisited. Neuroscience & Biobehavioral Reviews, 72, 95-110.

Method:

5) With regard the physical-to-virtual movement gain, there is mention of movement on the table and display being the same, but that can’t be possible. The movement on the table was only ~27 cm, while it was projected onto a large projector nearing a 2-m vertical amplitude (i.e., movement on the display will have gone visually further and faster than the limb in reality).

6) The following description does not seem to compute: “The velocity at which the target moved was 145 cm/s. A 4 × 6 cm target moving at 145 cm/s and a 2 × 4 cm effector resulted in a 68,96 ms time window” (Ln. 192-193). Is this 6,896 ms (6.896 secs)? If the target is travelling 213 cm at 145 cm/s, then it ought to be nearer ~1.5 secs of travel time.

7) The following description could be more clearly worded: “After 10 min of reaching the performance criterion started the Exposure phase” (Ln. 233-234).

8) While the rationale for only 9 trials per category of perturbation is robust within the response letter, we’re none-the-wiser when it comes to reading solely the manuscript. Some of the logic explained (including Fonseca et al. citation) should be highlight here (Ln. 236).

9) It states “The time window of Pfast was 50 ms, and the time window of Pslow was 111 ms” (Ln. 239), although if the onset of the target velocity perturbation was contingent upon the onset of the interceptive movement (with varying time to initiate and velocity of movement), then should the “time window” not between trials.

10) Upon second reading of the calculus of error (Ln. 206-207, 254-255), I’m unsure exactly about how it was really done. The description in Ln. 206-207 would make me think it was the distance between the target (yellow) and effector (green) at the moment of the former reaching the “strike zone”. However, viewing Figure 3 would make me think it was all about the distance between the target and “strike zone” at the moment that the effector reached the “strike zone”. Please explain and make clear the implication of direction (i.e., negatively signed scores indicate too slow an interceptive movement, while positively signed scores indicate a pre-exempted response).

11) The explanation offered for the “Pre” and “Post” trials is not entirely clear (Ln. 264-268). For example, the following statement on “The Pre trials are the reference to identify what happens with the performance and with the motor control when perturbations are inserted” reads like it is the perturbation trials themselves (thus bearing no distinction from “P trials”).

12) The detail behind the perturbation blocks also gets somewhat muddled (Ln. 271-273). There has to be an easier way to explain or simplify this.

**Do you want your identity to be public for this peer review?** For information about this choice, including consent withdrawal, please see our Privacy Policy

Reviewer #1: No

Reviewer #2: No

---

## [Author Response · Author response to Decision Letter 2]

13 Jun 2025

Response to Reviewer 2

We sincerely thank the reviewers again for their news insightful comments and constructive suggestions, which have significantly contributed to enhancing the clarity and overall quality of the manuscript. All modifications made to the manuscript are indicated in blue.

Major comments

Introduction:

1) While stabilization has been operationally defined within the manuscript, I find the explanation offered in the response letter to be more informative. Perhaps include some of this content, including details of empirical findings (e.g., Campos et al., 2022), in order to better envisage the state of (super-)stabilization.

We completely agreed that including empirical details would facilitate a better understanding of the state of (super-)stabilization. So, we incorporated this information between lines 68 and 114 (blue lines).

Method

2) I remain unconvinced by the failure to incorporate “exposure” into the analysis. It has been explained within the response letter that changes/learning were not of primarily concern when it came to the perturbation trials, although it begs the question: why consider “exposure” in the first place including the separation of perturbation trial blocks (i.e., x3 trials per block)?

We considered exposure in the first place because our aim was to analyze the control and performance of two levels of performance stabilization in the face of unpredictable perturbations. We did not consider the first phase (learning phase) because we took as a starting point what other studies have already shown: the stabilization and superstabilization groups already end the learning phase differently, with different skills. The difference between the two groups has been shown in studies such as, for example, Ugrinowitsch et al. (2011); Fonseca et al. (2012); Ugrinowitsch et al. (2014); Santos et al. (2017); Couto et al. (2021). Therefore, we assumed that the groups are different and inserted them into a phase with 18 disturbances, nine fast and nine slow.

In the exposure phase, we chose to analyze our dependent variables in three blocks of three trials at the pre- and post-perturbation moments, and not in just one block with nine trials, because the first organization allows us to glimpse the effects of the independent variable from the beginning to the end of the exposure phase. More specifically, we can glimpse at different moments (which we understand as beginning, middle and end of exposure) throughout the 129 trials, when each group begins to modify their control strategies and at what moment these modifications become effective, that is, they influence performance in a positive way in the face of the perturbations.

If we had analyzed the trials in just one block, we would only be able to show that the groups deal differently with the perturbations and also that they are affected differently by them after they are removed (moment Post-P). However, this second analysis did not allow us to see, for example, that despite the superiority of the superstabilization group, at the beginning of the exposure, when the first PII were inserted, this group showed a control strategy (tPV%) and performance (CE) similar to the stabilization group. However, in the middle and end of the exposure, the superstabilization group showed an effective control strategy to maintain performance, which the stabilization group was not able to.

The figure below illustrates the performance of the stabilization and superstabilization groups in just one block with the nine PII. If we had used this analysis, we could interpret that the superstabilization group had already started the exposure phase superior to the stabilization group, which the analysis in blocks with three trials shows is not true.

Results

3) While the direction of effects has been made much clearer, the failure to decompose interactions within the text still remains. Instead, I’m having to solely observe the between-group differences purported in figures 3 and 4 (via symbols) because the text mostly describes common differences between blocks within SG and SSG groups (synonymous with a block main effect) (Ln. 312-315, 327-330). When it comes to the interaction, I would advise focusing the report on the differences highlighted within the fore mentioned figures (e.g., CE in fast speed perturbation: SSG < SG for P, SSG > SG for Post).

We appreciate the suggestion and rewrite the results of the interactions in the lines that there are in blue color between lines 351 and 435.

Minor comments

Introduction

4) I may have misquoted the year of citation for Elliott et al. Here is the reference:

Elliott et al. (2017). The multiple process model of goal-directed reaching revisited. Neuroscience & Biobehavioral Reviews, 72, 95-110.

Thak you by the reference.

Method

5) With regard the physical-to-virtual movement gain, there is mention of movement on the table and display being the same, but that can’t be possible. The movement on the table was only ~27 cm, while it was projected onto a large projector nearing a 2-m vertical amplitude (i.e., movement on the display will have gone visually further and faster than the limb in reality).

Yes, the vertical projector had a total length of 228 cm, but we configured the virtual effector image to measure only 27 cm, as illustrated in the figure 1. We have made adjustments to the size of the effector rail in figure 1 in a way that it is more proportional to the actual size. Thank you for your attention with our manuscript.

6) The following description does not seem to compute: “The velocity at which the target moved was 145 cm/s. A 4 × 6 cm target moving at 145 cm/s and a 2 × 4 cm effector resulted in a 68,96 ms time window” (Ln. 192-193). Is this 6,896 ms (6.896 secs)? If the target is travelling 213 cm at 145 cm/s, then it thought to be nearer ~1.5 secs of travel time.

We understand your question, and we realized that the definition of time window was not well described in line 232, which caused this doubt. We improved the writing (lines 233 and 234) to make it clearer. The time window does not refer to the travel time, but rather to the time in which the target is within the strike zone and can be contacted by the effector.

7) The following description could be more clearly worded: “After 10 min of reaching the performance criterion started the Exposure phase” (Ln. 233-234).

Thank you by the suggestion. We appreciate it and incorporated it in lines 273 and 274.

8) While the rationale for only 9 trials per category of perturbation is robust within the response letter, we’re none-the-wiser when it comes to reading solely the manuscript. Some of the logic explained (including Fonseca et al. citation) should be highlight here (Ln. 236).

Thank you for the suggestion. We appreciate it and made the change between the lines 277 and 281.

9) It states “The time window of Pfast was 50 ms, and the time window of Pslow was 111 ms” (Ln. 239), although if the onset of the target velocity perturbation was contingent upon the onset of the interceptive movement (with varying time to initiate and velocity of movement), then should the “time window” not between trials.

We understand that the definition of time window was not well described in line 224, which caused this doubt. Considering the corrected form in the writing (lines 233 and 234), the change in the target's speed will cause a change in the possible time of contact between the effector and the target in the strike zone (time window). Therefore, we describe the changes in the time windows caused by the perturbations.

10) Upon second reading of the calculus of error (Ln. 206-207, 254-255), I’m unsure exactly about how it was really done. The description in Ln. 206-207 would make me think it was the distance between the target (yellow) and effector (green) at the moment of the former reaching the “strike zone”. However, viewing Figure 3 would make me think it was all about the distance between the target and “strike zone” at the moment that the effector reached the “strike zone”. Please explain and make clear the implication of direction (i.e., negatively signed scores indicate too slow an interceptive movement, while positively signed scores indicate a pre-exempted response).

We actually understood that the description of how the constant error was obtained was not clear. Moreover, your understanding from the figure is correct. We insert the information about the constant error in lines 297-299.

11) The explanation offered for the “Pre” and “Post” trials is not entirely clear (Ln. 264-268). For example, the following statement on “The Pre trials are the reference to identify what happens with the performance and with the motor control when perturbations are inserted” reads like it is the perturbation trials themselves (thus bearing no distinction from “P trials”).

We improved the wording to make it clearer between lines 308 and 310. Thank you for the suggestion.

12) The detail behind the perturbation blocks also gets somewhat muddled (Ln. 271-273). There has to be an easier way to explain or simplify this.

Thank you for the observation. We a change the write between the lines 310 and 315; 319 and 321.

---

## [Decision Letter · Decision Letter 2]

22 Jul 2025

Dear Dr. Ugrinowitsch,

Thank you for submitting your manuscript to PLOS ONE. After careful consideration, we feel that it has merit but does not fully meet PLOS ONE’s publication criteria as it currently stands. Therefore, we invite you to submit a revised version of the manuscript that addresses the points raised during the review process.

We look forward to receiving your revised manuscript.

Kind regards,

Job Fransen

Academic Editor

PLOS ONE

Journal Requirements:

Additional Editor Comments :

Dear authors,

Reviewer two has made some really valuable suggestions, which which I concur. There is an overall lack of claroty throughout your manuscript concerning its methodology and statistical analyses. At this time, I am inclined to suggest a rejection if these issues cannot be rectified upon review. It seems pretty clear that your work has merit (one of the reviewers has accepted after your most recent amendments), but both myself and reviewer two have some lingering concerns.

Thank you for addressing these promptly.

Job

Reviewers' comments:

Reviewer's Responses to Questions

**Comments to the Author**

Reviewer #2: (No Response)

2. Is the manuscript technically sound, and do the data support the conclusions?

Reviewer #2: Yes

3. Has the statistical analysis been performed appropriately and rigorously?

Reviewer #2: No

4. Have the authors made all data underlying the findings in their manuscript fully available?

Reviewer #2: Yes

5. Is the manuscript presented in an intelligible fashion and written in standard English?

Reviewer #2: Yes

Reviewer #2: General comments: The authors have entirely addressed previous comments surrounding the study rationale and supporting evidence within the Introduction. However, my uncertainly grows around the description of the analysis and eventual statistical outcomes. See below for details.

Major Comments:

1) The explanation of the trials does not entirely make sense, and what’s more, it may not even help matters. Either remove or elaborate (Ln. 310-315).

2) The difficulty in understanding may be the distinction between what is effectively x3 levels of exposure (pre, intermediate, post-) and x3 levels of block (trials 1-3, 4-6, 7-9). That is, there appears a confluence in the explanation for each of these factors (Ln. 316-321). Perhaps more definitively distinguish each of these (e.g., …exposure pertained to the immediately before (Pre), during (P) and after (Post) a perturbation, which was further brokered into block 1 (trials 1-3), block 2 (trials 4-6) and block 3 (trials 7-9).

3) The nomenclature used to define exposure and block is not exactly intuitive or easy to follow (i.e., “Block 1 with Pre, P1fast and Post, b) Block 2 with Pre, P2fast and Post…”) (Ln. 321-323). How about something like the following instead?: Pre1-3fast/slow, P1-3fast/slow, Post1-3fast/slow

4) Fore mentioned confusion is further compounded by the description of the two-way ANOVA where it appears block (1-3) was featured as a factor, but in reality it was exposure (Pre, P, Post) (Ln. 327-329).

5) (Unless I’m missing something completely here!) The same problem applies for the description of the inferential statistics (i.e., there is a constant reference to ‘blocks’ when I think it should be ‘exposure’, while blocks [1-3] were assessed separately in different sub-sections of the Results) (Pg. 14-18).

6) While pairwise difference have been more explicitly incorporated, they fail to decompose the group x block (i.e., exposure) interaction (e.g., “Post hoc analysis revealed that the same pattern was observed for the GS (p < 0.05)”). That is, the inferential stats report a similar pattern of influence of exposure within SSG and SG; thus, there would seem no difference between groups as a function of exposure. Meanwhile, there are asterisks within Figure 3 and 4 alluding to so-called “Interaction differences”. Due to this uncertainly we cannot really comprehend what has really been found here, and thus we cannot conclude anything concrete.

Minor comments:

7) “Consecutives” should be “consecutive” (i.e., no “s”) (Ln. 92).

8) “Internal Models” is capitalised either in error or for some unknown reason.

9) Excessively long sentence with some missing connecting words and/or punctuation (Ln. 115-120).

10) “GS” typos instead of “SG” (e.g., Ln. 419).

11) Tendency to mix up the past and present terms for ‘withdraw’ (e.g., “withdrawal” instead of “withdrawn”).

**Do you want your identity to be public for this peer review?** For information about this choice, including consent withdrawal, please see our Privacy Policy

Reviewer #2: **Yes: ** James Roberts

---

## [Author Response · Author response to Decision Letter 3]

2 Sep 2025

Response to the Editor

We sincerely thank the Editor and Reviewer 2 for their attention to our manuscript and for the opportunity to receive important feedback to improve the quality of our final product.

We reorganized the variable names to make them consistent with the data analyses, as Reviewer 2 identified coherence issues. All modifications made to the manuscript are indicated in green and have been incorporated into the graphs.

In the Measures and Data Analysis section, we modified the order of the dependent variables description (lines 296–305) to maintain consistency with the order of presentation in the Results, which was: number of corrections, relative time to peak velocity, and constant error. In the previous version of the manuscript, these variables had been presented in the following order: constant error, time to peak velocity, and number of corrections.

Response to Reviewer 2

We sincerely thank the reviewer 2 once again for insightful comments and constructive suggestions. We feel privileged to receive such valuable feedback, which has significantly contributed to improving the clarity of our manuscript.

All modifications made to the manuscript are indicated in green.

Major Comments:

1) The explanation of the trials does not entirely make sense, and what’s more, it may not even help matters. Either remove or elaborate (Ln. 310-315).

Thank you for the suggestion. We removed it, and the paragraph is clearer now (see topic Measures and data analysis).

2) The difficulty in understanding may be the distinction between what is effectively x3 levels of exposure (pre, intermediate, post-) and x3 levels of block (trials 1-3, 4-6, 7-9). That is, there appears a confluence in the explanation for each of these factors (Ln. 316-321). Perhaps more definitively distinguish each of these (e.g., …exposure pertained to the immediately before (Pre), during (P) and after (Post) a perturbation, which was further brokered into block 1 (trials 1-3), block 2 (trials 4-6) and block 3 (trials 7-9).

Thank you for your observation. You are correct; there was indeed some overlap in the explanation regarding blocks and moments. We reorganized the variable names to ensure they are consistent with data analyses.

First, we would like to clarify that “exposure” is not the moment with the Perturbation. The “exposure” refers to the entire experimental phase, comprising 129 trials. The exposure phase includes111 control trials, 9 fast perturbation trials (Pfast), and 9 slow perturbation trials (Pslow). We attempted to make this clear in lines 274 and 275 of the revised manuscript.

Regarding our confluence of the factors, to improve clarity, we revised the text between lines 312 and 337, as well as in the description of the results (lines 346-451). Each type of perturbation (i.e., fast and slow) was organized into three “moments”: Early (trials 1-3), Intermediate (trials 4-6), and Later (trials 7-9). Each “moment” was divided into three blocks — Pre, P (perturbation), and Post — and each block represents the average of three trials, which means we organized each perturbation trial with its respective surrounding control trials (Pre and Post). This organization was applied separately to fast and slow perturbations.

3) The nomenclature used to define exposure and block is not exactly intuitive or easy to follow (i.e., “Block 1 with Pre, P1fast and Post, b) Block 2 with Pre, P2fast and Post…”) (Ln. 321-323). How about something like the following instead?: Pre1-3fast/slow, P1-3fast/slow, Post1-3fast/slow.

This question is closely linked to Question 2, and we likewise acknowledge that the original description may have led to confusion. We believe that the changes made in lines 311–338 and 347, 360, 378,402, 415 and 436 clarify the nomenclature problem.

4) Fore mentioned confusion is further compounded by the description of the two-way ANOVA where it appears block (1-3) was featured as a factor, but in reality it was exposure (Pre, P, Post) (Ln. 327-329).

Your observation is correct. As we cited before, the confusion stemmed from an imprecise description of the relationship between blocks and moments. We reorganized the whole manuscript based on the answer to comment #2. Summarizing

- Early Moment: trials 1–3 with perturbation, along with their corresponding Pre and Post trials;

- Intermediate Moment: trials 4–6 with perturbation and their respective Pre and Post trials;

- Later Moment: trials 7–9 with perturbation and their corresponding Pre and Post trials.

Two-way ANOVA was conducted between the two groups to compare the following variables: N-Cor, tPV%, and CE across the three blocks (Pre, P, and Post), separated by moment (Early, Intermediate and Later) and type of perturbation, Pslow and Pfast.

We have modified the text between lines 336 and 338 to improve clarity and address the reviewer’s comment.

5) (Unless I’m missing something completely here!) The same problem applies for the description of the inferential statistics (i.e., there is a constant reference to ‘blocks’ when I think it should be ‘exposure’, while blocks [1-3] were assessed separately in different sub-sections of the Results) (Pg. 14-18).

You are correct, the original description was unclear. We believe we have addressed this issue with the revised explanation provided between lines 313 and 338, as well as with the updated symbols of figures 3 and 4. The statistical description is now correct and consistent with the definitions of blocks and moments presented in that section.

6) While pairwise difference have been more explicitly incorporated, they fail to decompose the group x block (i.e., exposure) interaction (e.g., “Post hoc analysis revealed that the same pattern was observed for the GS (p < 0.05)”). That is, the inferential stats report a similar pattern of influence of exposure within SSG and SG; thus, there would seem no difference between groups as a function of exposure. Meanwhile, there are asterisks within Figure 3 and 4 alluding to so-called “Interaction differences”. Due to this uncertainly we cannot really comprehend what has really been found here, and thus we cannot conclude anything concrete.

The same behavioral pattern was observed for SG and SSG only in relation to the behavioral tendency towards control and perturbation trials. For example, when the perturbation was introduced, the CE of both groups increased compared to the control trials (Pre block), in both Pslow and Pfast conditions. Despite the increase, for example, in the Last Moment of PSlow, the GSS showed a lower constant error than SG in the face of perturbations. Subsequently, when the perturbation was removed, the constant error of both groups decreased in the control trials (Post) compared to the perturbation trials (P) in both fast and slow perturbations. This description elucidates the same behavioral pattern (increase (in P) and decrease (in Post) of error) for SG and SSG, but with SG being superior only on Intermediate and Last Moments of Slow Perturbation. This is because this group, despite having an increased error in P compared to the Pre, presented a lower constant error than SG when faced with the Perturbation slow (P4-6 slow and P7-9slow).

More specifically, in the decomposition of interaction groups x blocks, a similar pattern of behavior between SG and SSG was seen in the Intermediate and Last moments of the fast perturbation, but without superiority of SG, as shown below and described in detail between lines 367-373 and 383-389 of the manuscript.

1) Intermediate Moment: Pre x P4-6fast x Post

Dependent variable: Constant Error (CE)

Description: In relation to control trials (Pre), the CE increased in both groups when the perturbation was inserted (P4-6fast), then when the perturbation was withdrawn (Post), the CE decreased in relation to P4-6fast. Specifically, in this moment (Intermediate Moment), for fast perturbations, the SSG is not superior to the SG. Therefore, both groups showed a similar behavior pattern.

2) Last Moment: Pre x P7-9fast x Post

Dependent variable: Constante Error (CE)

Description: In relationship to control trials (Pre), the CE increased in both groups when the perturbation was inserted (P7-9fast), then when the perturbation was withdrawn (Post), the CE decreased in relation to P7-9fast. Specifically in this moment (Intermediate Moment), to fast perturbation, the SSG it is not superior to the SG. Therefore, both groups showed a similar behavior pattern.

Regarding the slow perturbation, although SG and SSG showed a similar pattern of behavior at the Intermediate and Last moments, SSG was superior to SG. The behavior patterns are presented below and described in detail between lines 428–431 and 444–457 of the manuscript.

1) Intermediate Moment: Pre x P4-6slow x Post

Dependent variable: Constante Error (CE)

Description: In relationship to control trials (Pre), the CE increased in both groups when the perturbation was inserted (P4-6slow), then when the perturbation was withdrawn (Post), the CE decreased in relation to P trials. It is important to highlight that in this moment (Intermediate Moment), SSG showed lower CE than SG in P4-6slow, i.e., when Pslow was inserted. Furthermore, the SSG superiority can be observed by comparing the CE in the perturbation block with the previous control trials (Pre). This comparison shows that the SSG, unlike the SG, was able to maintain its performance (CE) from the previous control trials (Pre) even after the perturbation was inserted. In contrast, SG increased the CE in response to the perturbation when compared to its performance in the previous control trials (Pre).

2) Intermediate Moment: Pre x P7-9slow x Post

Dependent variable: Constante Error (CE)

Description: In relationship to previous control trials (Pre), the CE increased to both groups when the perturbation was inserted (P7-9slow), then when the perturbation was withdrawn (Post), the CE decreased in relationship P7-9slow. It is important highlight that in this moment (Intermediate Moment), the SSG showed lower CE than SG in P7-9slow, i.e. when Pslow was inserted. Furthermore, SSG superiority can be observed by comparing the CE in the perturbation block with the block of previous control trials (Pre). This comparison shows that the SSG, unlike the SG, was able to maintain its performance (CE) from the previous control trials (Pre) even after the perturbation was inserted. In contrast, SG increased the CE in response to the perturbation when compared to its performance in the block of previous control trials (Pre). Finally, in this moment (Last Moment), it was found difference between groups. The post-hoc showed that SSG had lower CE than SG.

The superiority of SSG's performance can be understood from the analyses of N-cor and tPV%. The results of these motor control variables showed that, at the "Intermediate" (P4-6 slow) and "Last" (P7-9 slow) moments, when perturbations were introduced, SSG presented a lower tPV% and performed more corrections than SG. These results indicate that, when the perturbation is slow, the group that achieves superstabilization of performance is more competent in using the strategy of reducing the time to peak velocity, which allows make corrections in the Last portions of the movement. Furthermore, SSG performed a higher number of corrections than SG, which were also more effective, during Pslow. These results allow for the maintenance of the previous performance demonstrated Pre trials, and also have a superior performance than SG, which also performed corrections, but not as effectively as SSG.

We updated the symbols representing the results of the interactions and updated the captions in lines 397-398 and 460-461.

Minor comments

7) “Consecutives” should be “consecutive” (i.e., no “s”) (Ln. 92).

Thank you for the observation. We made the correction on line 92.

8) “Internal Models” is capitalised either in error or for some unknown reason.

Thank you by observation. We made the correction on line 118.

9) Excessively long sentence with some missing connecting words and/or punctuation (Ln. 115-120).

Thank you for the observation. We inserted a full stop on line 118.

10) “GS” typos instead of “SG” (e.g., Ln. 419).

Thank you for the observation. We made the correction on lines 371, 387, 430 and 449.

11) Tendency to mix up the past and present terms for ‘withdraw’ (e.g., “withdrawal” instead of “withdrawn”).

Thank you by observation. We made the correction on lines 354, 369, 372, 386, 388, 409, 423, 428, 448 and 452.

---

## [Decision Letter · Decision Letter 3]

30 Sep 2025

Dear Dr. Ugrinowitsch,

We look forward to receiving your revised manuscript.

Kind regards,

Dimitris Voudouris

Academic Editor

PLOS ONE

Journal Requirements:

Reviewers' comments:

Reviewer's Responses to Questions

**Comments to the Author**

Reviewer #2: All comments have been addressed

2. Is the manuscript technically sound, and do the data support the conclusions?

Reviewer #2: Yes

3. Has the statistical analysis been performed appropriately and rigorously?

Reviewer #2: Yes

4. Have the authors made all data underlying the findings in their manuscript fully available?

Reviewer #2: Yes

5. Is the manuscript presented in an intelligible fashion and written in standard English?

Reviewer #2: Yes

Reviewer #2: General comments:

The authors had adequately addressed any previous comments; especially, the outline of study factors and subsequent pairwise comparisons. I have but a few minor comments to tease apart or elaborate on the previously made changes. Assuming these minor changes are implemented, then I would recommend acceptance for publication.

Minor comments:

1) The modified description of the analysis could be made clearer and more explicit (Ln. 335-337). Perhaps the following: “The CE and tPV% measures from each of the exposure phase moments (Early, Intermediate and Later) and perturbations (fast, slow) were analysed using a two-way mixed-design ANOVA consisting of group (SG and SSG) and block (Pre, P, Post) factors.”

2) Post hoc analysis was unable to decouple the interaction between group and block for the following: Intermediate moment: Pre P x P4-6fast x Post P (Ln. 366-372), Later moment: Pre P x P7-9fast x Post P (Ln. 382-388). This should be made clearer within the text report (along similar lines to that reported by the authors in the previous ‘response to comments’ letter).

3) Potential misreport in the post hoc that decouples the interaction (Ln. 430): “…the SSG showed lower (closest to zero) CE than SG in P4-6slow and similar CE in Pre and P4-6slow (p < 0.05).”

**Do you want your identity to be public for this peer review?** For information about this choice, including consent withdrawal, please see our Privacy Policy

Reviewer #2: **Yes: ** James W. Roberts

---

## [Author Response · Author response to Decision Letter 4]

6 Oct 2025

We sincerely thank Reviewer 2 for the attention to our manuscript. All modifications made to the manuscript are indicated in green.

Minor comments:

1) The modified description of the analysis could be made clearer and more explicit (Ln. 335-337). Perhaps the following: “The CE and tPV% measures from each of the exposure phase moments (Early, Intermediate and Later) and perturbations (fast, slow) were analyzed using a two-way mixed-design ANOVA consisting of group (SG and SSG) and block (Pre, P, Post) factors.”

We have made the modification as suggested between lines 335 and 337.

2) Post hoc analysis was unable to decouple the interaction between group and block for the following: Intermediate moment: Pre-P x P4-6fast x Post P (Ln. 366-372), Later moment: Pre-P x P7-9fast x Post P (Ln. 382-388). This should be made clearer within the text report (along similar lines to that reported by the authors in the previous ‘response to comments’ letter).

We have rewritten it as suggested between lines 369–375 and 387–393.

3) Potential misreport in the post hoc that decouples the interaction (Ln. 430): “…the SSG showed lower (closest to zero) CE than SG in P4-6slow and similar CE in Pre and P4-6slow (p < 0.05).”

Indeed, P4-6slow was a misreport. Regarding CE, the similarity between groups at the intermediate moment occurred in the Pre and Post perturbation phases, but not during the perturbation (P4-6slow). We have made the correction in line 435.

---

## [Decision Letter · Decision Letter 4]

7 Oct 2025

Practice beyond performance stabilization increases the use of online adjustments to unpredictable perturbations in an interceptive task

PONE-D-24-43982R4

Dear Dr. Ugrinowitsch,

We’re pleased to inform you that your manuscript has been judged scientifically suitable for publication and will be formally accepted for publication once it meets all outstanding technical requirements.

Kind regards,

Dimitris Voudouris

Academic Editor

PLOS ONE

Additional Editor Comments (optional):

Reviewers' comments:

Reviewer's Responses to Questions

**Comments to the Author**

Reviewer #2: All comments have been addressed

2. Is the manuscript technically sound, and do the data support the conclusions?

Reviewer #2: Yes

3. Has the statistical analysis been performed appropriately and rigorously?

Reviewer #2: Yes

4. Have the authors made all data underlying the findings in their manuscript fully available?

Reviewer #2: Yes

5. Is the manuscript presented in an intelligible fashion and written in standard English?

Reviewer #2: Yes

Reviewer #2: (No Response)

**Do you want your identity to be public for this peer review?** For information about this choice, including consent withdrawal, please see our Privacy Policy

Reviewer #2: **Yes: ** James W. Roberts

---

## [Editor Report · Acceptance letter]

PONE-D-24-43982R4

PLOS ONE

Dear Dr. Ugrinowitsch,

I'm pleased to inform you that your manuscript has been deemed suitable for publication in PLOS ONE. Congratulations! Your manuscript is now being handed over to our production team.

Kind regards,

on behalf of

Dr. Dimitris Voudouris

Academic Editor

PLOS ONE